# BIGOS V2 Benchmark for Polish ASR: Curated Datasets and Tools for Reproducible Evaluation

Michał Junczyk[1, 2]

[1]Adam Mickiewicz University, Poznań, Poland, michal.junczyk@amu.edu.pl
[2]Allegro, Poznań, Poland, michal.junczyk@allegro.com

## Abstract

Speech datasets available in the public domain are often underutilized because of challenges in accessibility and interoperability. To address this, a system to survey, catalog, and curate existing speech datasets was developed, enabling reproducible evaluation of automatic speech recognition (ASR) systems. The system was applied to curate over 24 datasets and evaluate 25 ASR models, with a specific focus on Polish. This research represents the most extensive comparison to date of commercial and free ASR systems for the Polish language, drawing insights from 600 system-model-test set evaluations across 8 analysis scenarios. Curated datasets and benchmark results are available publicly. [1] The evaluation tools are open-sourced to support reproducibility of the benchmark, encourage community-driven improvements, and facilitate adaptation for other languages.[2]

## 1 Introduction

### 1.1 Background

The Polish language is spoken by more than 50 million people worldwide. The number of available ASR systems and services, as well as speech data resources that support Polish, is systematically growing. However, the community has insufficient resources to methodically evaluate and track progress of ASR (Automatic Speech Recognition) technology for Polish. First, the available data assets are underutilized due to challenges such as accessibility, licensing, and interoperability. Secondly, there is no standardized ASR benchmark dataset. Finally, the tooling to reproduce or systematically extend evaluation scope is missing. As a result Polish ASR systems benchmarks performed so far cover limited number systems and datasets (see Appendix B.1). [15, 27, 28, 41, 17, 10] These limitations may slow the development of new systems and applications, as reliable evaluations and publicly available *leaderboards* drive research progress and inform the public about the capabilities of AI technology. [22] The international ASR community has recognized the importance of evaluation methodologies for consistent and comparative performance assessments in ASR specifically. [2, 38, 7] and ML field in general [19, 24, 23] This calls for innovations in the management of ASR datasets and evaluation methods. [13, 40]

### 1.2 Research gap

Current data curation and ASR benchmarking methods for low-resource languages such as Polish exhibit several shortcomings:

---

[1]BIGOS collection on Hugging Face
[2]BIGOS ASR eval tools on GitHub

38th Conference on Neural Information Processing Systems (NeurIPS 2024) Track on Datasets and Benchmarks.

- **Data utilization:** Speech datasets are often underutilized due to limited awareness or restricted accessibility.

- **Data quality:** Insufficient understanding of test sets can lead to an inaccurate representation of state-of-the-art performance.

- **Evaluation reproducibility:** Limited adoption of common benchmark sets impedes the validation and replication of research results.

- **Evaluation scope:** Ecologically valid ASR evaluations require consideration of a broader range of datasets, systems, and performance metrics to ensure comprehensive assessment.

## 1.3 Contributions

1. **Benchmark datasets curation:** To address the lack of standardized ASR evaluation resources for Polish, a benchmark dataset was curated from 24 openly available sources.[3] Diverse samples of both read[4] and spontaneous speech[5] are included.

2. **Benchmark toolchain development:** A benchmark toolchain was developed to ensure consistent ASR evaluation through standardized protocols, with flexible support for incorporating new datasets, systems, and metrics.[6]

3. **ASR systems evaluation:** Using the curated dataset, nine ASR systems and twenty-five models, including both commercial and freely available solutions, were evaluated. Variations in performance across different systems, datasets, and speaker demographics were observed. Results are available publicly on the Polish ASR leaderboard.[7]

4. **Open resources sharing:** All datasets, tools, and evaluation results are made freely available to the research community. This promotes transparency, reproducibility, and collaboration, allowing researchers to leverage the resources for further Polish ASR development or adapt them to other languages.

## 2 Methodology

### 2.1 System overview

The system developed for data curation and ASR benchmarking encompasses three main processes:

- **ASR speech datasets survey**: Involves analyzing speech data catalogs and taxonomies, creating a dashboard that summarizes and categorizes existing speech datasets.

- **Curation of ASR benchmark dataset**: Includes processing, formatting, and analyzing datasets to create a standardized set for benchmarking ASR systems. BIGOS (Benchmark Intended Grouping of Open Speech) format was used. [10]

- **Evaluation of ASR systems**: Involves managing the evaluation process, generating results, and presenting performance metrics through a public dashboard for system comparison and analysis.

Figure 1 illustrates the system architecture and the core open tools used for development. The subsequent sections provide a detailed description of the specific processes and tools.

---

[3]BIGOS datasheets

[4]BIGOS V2 dataset on Hugging Face

[5]PELCRA for BIGOS dataset on Hugging Face

[6]BIGOS ASR eval tools on GitHub

[7]AMU Polish ASR leaderboard on Hugging Face

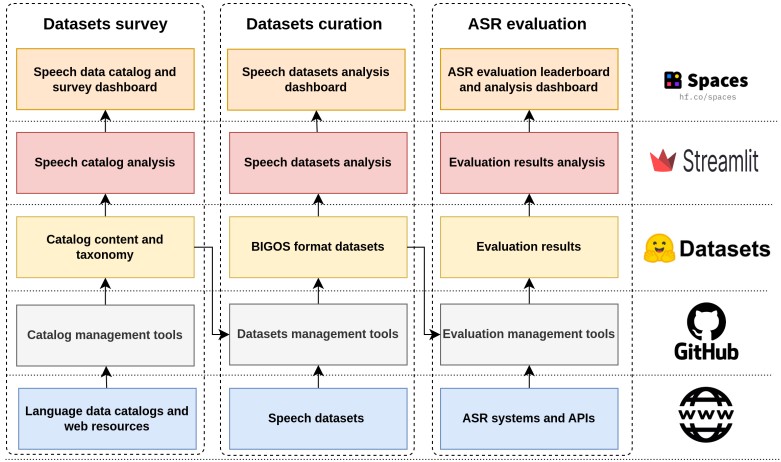

Figure 1: Architecture of data curation and ASR evaluation system.

## 2.2 Survey of datasets

A keyword-based literature review was used to identify and document relevant datasets. [34] The datasets were manually analyzed and annotated. The final methodology included:

1. Conducting keyword searches in relevant sources
2. Manually analyzing and annotating documentation
3. Cross-checking multiple sources for consistency and accuracy
4. Validating and analyzing downloadable datasets
5. Analyzing metadata to derive insights on Polish ASR speech datasets
6. Making the catalog and insights publicly available

The survey sources include language data repositories, scientific community platforms, and public domain documentation. The attributes considered include creator, funding, license, publication date, quality assurance, and content characteristics such as the format of the audio file and the number of speakers. [12] Resulting catalog and survey insights are shared on GitHub[8] and Hugging Face.[9]

## 2.3 Dataset curation

### 2.3.1 Design considerations

A curated benchmark dataset for Polish ASR systems is intended to have the following features:

- **Task-appropriate:** Relevant and practical for the intended ASR task.
- **Accessible:** Available online under a license allowing the free use and derivative works.
- **Discoverable:** Easy to find and acquire (no registration or other access barriers).
- **Diverse and challenging:** Containing various examples to test the adaptability of the model, as well as complex cases to encourage community participation and minimize the risk of benchmark saturation.
- **Annotated**: With metadata about speakers and recordings allowing nuanced analysis and interpretation of the results.
- **Optimally sized:** Large enough to be representative, but manageable to download and use.
- **Clean yet realistic:** Free of major errors, but noisy enough to represent the complexity of the real world.

---

[8]Polish ASR speech data survey on GitHub
[9]Polish ASR survey on Hugging Face

- **Well-documented:** Provided with documentation that is understandable to users without technical skills.
- **Well-explained:** Provided with evaluation baselines and how-to-use script examples.

### 2.3.2 Leveraging speech data catalog for sourcing open datasets

The Polish ASR speech dataset catalog was used to select datasets for curation. [11] Following criteria were considered:

- Datasets are available online under a license allowing free use for noncommercial purposes.
- Transcriptions are aligned with the recordings.
- Recording sampling rate is at least 8 kHz.
- Audio files are encoded using at least 16 bits per sample.

Twenty-four source datasets were curated as two new datasets: *BIGOS V2* and *PELCRA for BIGOS*. Named after the Polish dish *bigos*, a traditional cabbage-based stew — **BIGOS V2** builds upon its predecessor, *BIGOS (Benchmark Intended Grouping of Open Speech)* [10][10] and offers expanded selection of metadata and recordings from the following source corpora:

- **The Common Voice dataset** *(mozilla-common_voice_15-23)* [3] covers over 60 languages and many underrepresented groups. Available under CC-0 license.
- **The Multilingual LibriSpeech (MLS) dataset** *(fair-mls-20)* is a large multilingual corpus made by Facebook AI Research (FAIR) [31]. Derived from audiobooks, it covers eight languages, with 44,000 hours of English and 6,000 hours for other languages. The Polish data includes 137 hours from 25 books by 16 speakers. Available under CC-BY license.
- **The Clarin Studio dataset** *(clarin-pjatk-studio-15)* by CLARIN-PL includes 13,802 short utterances (56 hours) from 554 sessions by 317 speakers. Each session has 20-31 audio files, all recorded in a studio for clear audio. Available under CC-BY-SA license.
- **The Clarin Mobile dataset** *(clarin-pjatk-mobile-15)* is a Polish speech corpus of read speech recorded on a telephone. It includes many speakers reading several dozen sentences and words with rare phonemes. Available under CC-BY-SA license.
- **The Jerzy Sas PWR datasets** (Politechnika Wrocławska) comprise three legacy sets of recordings available in the public domain:
  - Male speaker speech set *(pwr-maleset-unk)* – single male speaker recordings.
  - Utterances containing short words (*pwr-shortwords-unk*) – single-phoneme conjunctions and prepositions likely to be misrecognized.
  - Spoken commands as very important utterances (VIUs) *(pwr-viu-unk)* – editor control commands and domain-specific utterances.
- **The M-AI Labs Speech corpus** *(mailabs-19)* created from audiobooks as *MLS*. Intended for training speech recognition and synthesis systems in nine languages, with nearly a thousand hours of audio, including 53.5 hours for Polish. Available under proprietary license.
- **The AZON Read and Spontaneous Speech datasets** *(pwr-azon_spont-20, pwr-azon_read-20)* contain recordings from academic staff in the physical chemistry domain, including both supervised readings and unsupervised spontaneous recordings such as interviews and presentations. Available under a CC-BY-SA license.[11]

  Compared to predeccessor, *BIGOS V2* contains curated recordings and metadata from the following source corpora:

- **Google FLEURS** *(google-fleurs-22)* is a parallel speech benchmark dataset in 102 languages, based on the FLoRes-101 machine translation benchmark. [6] Hosted on Hugging Face[12] and available under a CC-BY license.

---

[10]BIGOS V1 dataset on Hugging Face
[11]AZON dataset homepage
[12]FLEURS dataset homepage

- **PolyAI Minds14** (*polyai-minds14-21*) is a dataset for training and evaluating intent recognition systems using spoken data. Covers spoken samples in the commercial e-banking domain in 14 language variations. [8] Hosted on Hugging Face[13] and available under a CC-BY license.

  Additionally, **PELCRA for BIGOS** dataset contains recordings and metadata from the following source corpora:

- **PolEval 22 Diabiz sample** (*ul-diabiz_poleval-22)* was used for a punctuation restoration task in the 2022 PolEval competition. It is a subset of the *DiaBiz homepage*[14] dialog corpus of phone-based customer–agent interactions by the PELCRA group of the University of Łódź. Available publicly under CC-BY-SA-NC-ND and curated for Polish ASR systems benchmarking purposes with the consent of the author.

- **SpokesMix**[15] is a corpus of conversational Polish by the PELCRA group. [26] It includes speech recordings and word-by-word transcriptions with non-speech events. Available under the CC-BY-NC-ND license and curated with permission of the authors.

- **SpokesBiz**[16] is a corpus of conversational Polish from the CLARIN-BIZ project, featuring over 650 hours of recordings from nearly 600 speakers. [28] Transcriptions are diarized and manually annotated. Includes eight diverse subsets, e.g. biographical interviews, job interviews, podcasts, and student presentations. Available under the CC-BY-NC-ND license and curated with the authors permission.

Datasheets of curated datasets can be found in Appendices C.9, C.10, C.11, C.12 and Hugging Face.[17]

### 2.3.3 Curation process

- **Dataset structure curation:**

  - Downloading and manually inspecting format and contents
  - Creating train/dev/test splits if not available
  - Assigning standard IDs to speakers and files

- **Audio file curation:**

  - Removal of invalid and duplicated audio files
  - Unifying audio format to WAV 16 bits/16 kHz
  - Normalizing audio amplitude to -3 dBFS
  - Splitting long audio files into shorter segments based on time-alignment annotations

- **Text files (transcripts and metadata) curation:**

  - Converting text encoding to UTF8
  - Extracting original transcription and removing redundant characters
  - Removal of audio files for utterances containing offensive content
  - Extracting and unifying metadata contents
  - Generating metadata from text and audio content
  - Saving in the standard tabular format

- **Dataset distribution**

  - Uploading to the HF dataset hub
  - Referencing the original license and authors in the README file

The resulting *BIGOS utterance data object* with a description of the standard metadata fields is available in Table 24 in the Appendix.

---

[13]Minds14 dataset homepage

[14]Diabiz dataset homepage

[15]SpokesMix dataset homepage

[16]SpokesBiz dataset homepage

[17]BIGOS datasheets

### 2.4 ASR evaluation

#### 2.4.1 System design considerations

Below is an overview of the main design considerations. Established tools and platforms for data management and evaluation were used whenever feasible (see Appendices C.1 and C.2 for details).

- **Metrics**: Support for well-established metrics for ASR evaluation.
- **Extensibility**: Easy integration of new datasets, normalization methods, metrics, and systems.
- **Availability**: Publicly accessible and intuitive presentation of results.
- **Comprehensiveness**: Performance analysis across scenarios, system parameters, and user groups.

#### 2.4.2 Overview of the evaluation process

The process is presented on figure 2. Currently four evaluation metrics are supported: Sentence Error Rate (SER), Word Error Rate (WER), Match Error Rate (MER) and Character Error Rate (CER). [21] The definitions are provided in Appendix C.3. The same pipeline was used to normalize both references and hypotheses. The impact of normalization is discussed in section 3.1 and normalization steps are described in Appendix C.5. Python scripts used for the evaluation are available on GitHub.[18]

In total twenty-five models of nine ASR systems were evaluated: Google STT V1 and V2, Azure STT, Whisper local and cloud, AssemblyAI, NeMo, MMS and Wav2Vec2. The references and details are available in the Appendix C.4. The complete list of evaluated systems and models is presented in table 17.

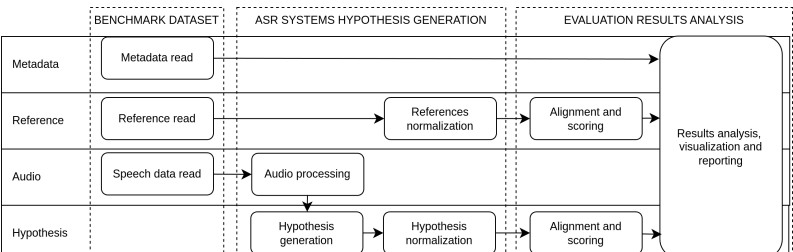

Figure 2: ASR evaluation process data flow.

## 3   Evaluation results

Eight evaluation scenarios encompassing several key dimensions are supported. The *All System Variants* scenario considers different system-model variants across the entire dataset, while *Subset Analysis* focuses on evaluating specific subsets of the test data. The *System Type Comparison* scenario contrasts free versus commercial systems, highlighting differences in performance. *Model Size Evaluation* assesses variants by their respective model sizes, and *Audio Duration Analysis* provides insight on the best and worst performing systems for different ranges of audio lengths. *Speaking Rate Evaluation* examines system performance across varying speech rates, while *Speaker Age Group Analysis* and *Speaker Gender Analysis* evaluate system variants based on speaker age and gender demographics, respectively.

All benchmark results can be accessed through the public interactive dashboard.[19] Users can display the evaluation results for a specific scenario or perform custom analysis for specific datasets, systems, metrics, normalization techniques, and diagram types. The results of selected scenarios are analyzed in the subsequent sections.

---

[18]BIGOS ASR eval tools on GitHub

[19]AMU Polish ASR leaderboard on Hugging Face

## 3.1 Impact of normalization on error rates

Table 1 presents the individual and average error rate reductions, measured in percentage points, for each normalization method applied. Corresponding results for the *PELCRA for BIGOS* dataset can be found in the Appendix C.6 and online.

Table 1: Reduction of error rates caused by normalization of references and hypothesis for *BIGOS V2 dataset*

| Method | SER [p.p.] | WER [p.p.] | MER [p.p.] | CER [p.p.] | Average [p.p.] |
|---|---|---|---|---|---|
| blanks | -1.79 | 0.00 | 0.00 | -0.85 | -0.66 |
| lowercase | -2.65 | -6.06 | -6.27 | -1.40 | -4.10 |
| punctuation | -1.40 | -7.61 | -7.95 | -1.67 | -4.66 |
| all | -24.90 | -14.63 | -15.22 | -4.04 | -14.70 |

## 3.2 Overall accuracy of available ASR systems and models

Figure 3 show the WER box plot for the systems evaluated using the *BIGOS V2 dataset*. The 3 best ASR models in terms of accuracy are *Whisper Large V3, Whisper Cloud* and *Assembly AI best*. Corresponding results for the *PELCRA for BIGOS* dataset can be found in the Appendix C.6 and online.

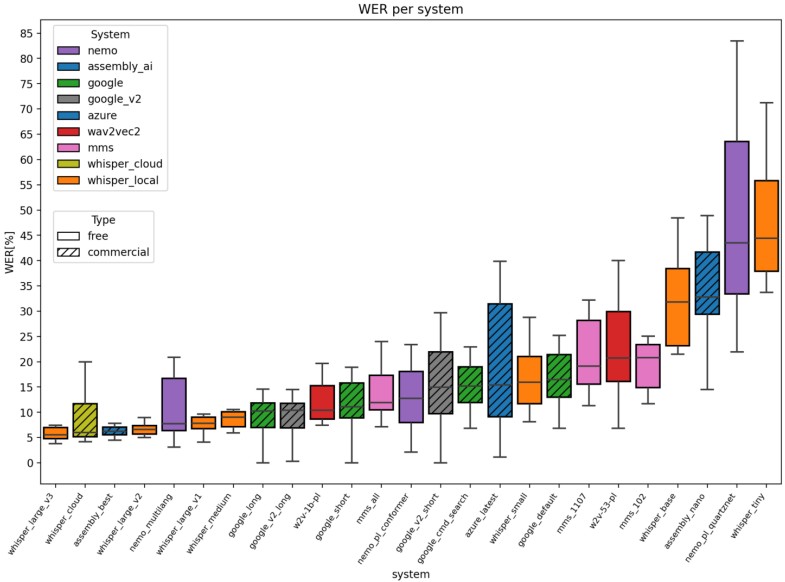

Figure 3: Box plot showing Word Error Rate (WER) distributions for systems evaluated on the *BIGOS V2 dataset*. Lower values indicate better performance, while narrower boxes and whisker ranges demonstrate more consistent performance across the 12 source datasets.

## 3.3 Subset analysis

Figure4 presents performance across subsets of the *BIGOS V2 dataset*, sorted by median WER. The *CommonVoice* and *PWR* subsets are the least challenging overall, though the *pwr-viu-unk* subset shows high WER for many systems. As revealed by manual inspection, this is caused by hallucinations for unnaturally slow speech rates. The most challenging subsets are *pwr-azon_read20*, *pwr-azon_spont20* and *polyai-minds14-21*, containing specialized terminology, spontaneous speech and varied accents, respectively . These factors contribute to increased difficulty for ASR systems, leading to significant performance variation across different models.

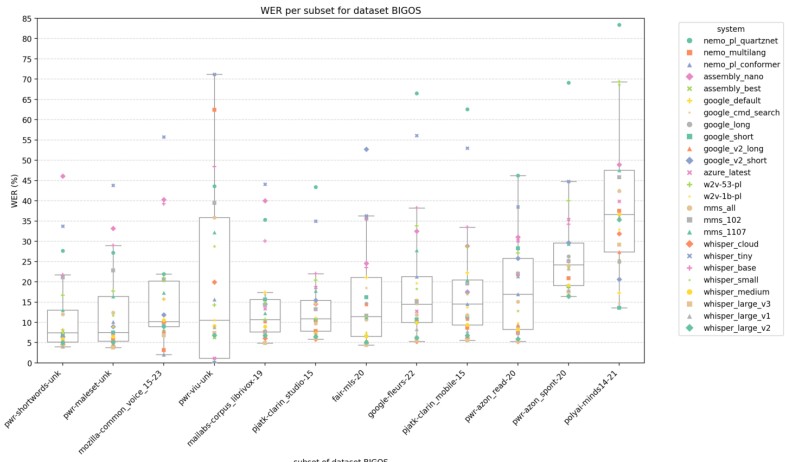

Figure 4: Boxplot of Word Error Rate (WER) per subset of *BIGOS V2 dataset*. Each box represents the WER distribution for a subset, with individual ASR systems indicated by unique colors and markers. Lower values indicate better performance.

## 3.4 Comparison of accuracy of commercial and freely available ASR systems

Table 2 compares the WER of commercial and free ASR systems. Commercial systems achieved lower minimum and median WER for BIGOS V2 and PELCRA datasets by approximately 2.5 p.p. and 4.5 p.p., respectively. Furthermore, both commercial and free systems obtained better recognition accuracy for read speech (BIGOS V2) than conversational speech (PELCRA) by approximately 17 and 19 p.p., respectively.

Table 2: WER statistics for freely available and commercial ASR systems.

| Dataset | Speech | Systems | Med. WER | Mean WER | Std. WER | Min. WER |
|---------|--------|---------|----------|----------|----------|----------|
| BIGOS V2 | read | paid | 12.96 | 17.26 | 24.98 | 0.00 |
| BIGOS V2 | read | free | 15.47 | 21.93 | 19.29 | 2.10 |
| PELCRA | spontaneous | paid | 29.90 | 31.34 | 14.72 | 5.27 |
| PELCRA | spontaneous | free | 34.18 | 37.45 | 19.43 | 8.74 |

## 3.5 Accuracy as a function of model size

Figures 5a and 6a present the relationship between model size and WER for BIGOS and PELCRA datasets, respectively. The figures show that as model size increases, WER generally decreases, indicating improved performance for larger models.

## 3.6 Accuracy as a function of speech rate

Figures 5b and 6b illustrate the relationship between WER and speech rate, defined as the average number of words spoken per second.

# 4 Discussion

## 4.1 Analysis of findings

### 4.1.1 Impact of normalization

Normalization techniques resulted in significant reductions in error rates for all types of metrics (SER, WER, MER, CER). Applying all methods reduced WER by 15.78 p.p. for the PELCRA dataset and

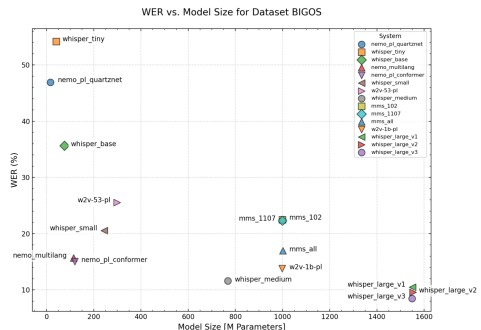
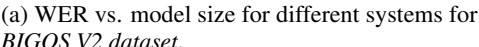

(a) WER vs. model size for different systems for *BIGOS V2 dataset*.

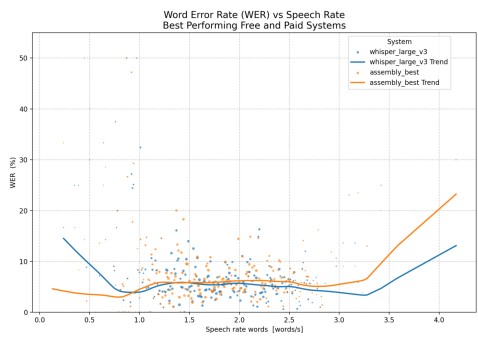

(b) WER as a function of speech rate for best performing free and paid systems for *BIGOS V2 dataset*.

Figure 5: Example of evaluation scenario results.

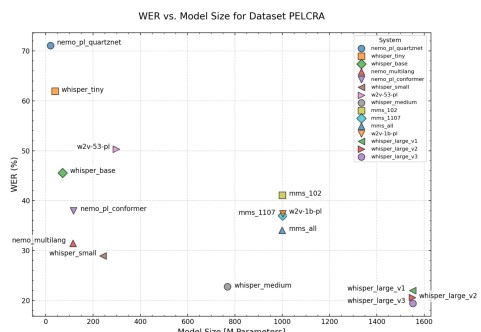

(a) WER vs. model size for different systems for *PELCRA for BIGOS* dataset.

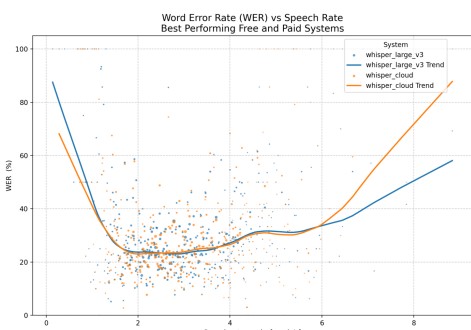

(b) WER as a function of speech rate for best performing free and paid systems for *PELCRA for BIGOS* dataset.

Figure 6: Example of evaluation scenario results.

15.22 p.p. for the *BIGOS V2 dataset*, highlighting the sensitivity of lexical metrics to spelling and formatting variations.

### 4.1.2 Determining the best systems among free and commercial

Conversational speech (PELCRA) has higher error rates due to its spontaneous nature, with greater variability in style, speed, and pauses. The read speech (BIGOS V2) contains more structured speech, resulting in lower WER.

### 4.1.3 Impact of model size on WER

Figure 5a shows that as the model size increases, the WER generally decreases, with larger models consistently achieving better performance. This trend is clear for models like the *Whisper* series, although there are significant variations between models of similar sizes, particularly those trained on different datasets, such as *MMS* and *Wav2vec2*. The *Whisper Large* models achieve the lowest overall WER, while smaller models such as *Nemo Multilang* and *Nemo Conformer* still manage competitive results relative to larger models, demonstrating efficiency.

Figure 6a shows that the WER in all models for the PELCRA data set is higher. The trend of decreasing WER with larger models holds for both datasets, but the gains from larger models are more pronounced for conversational, noisy speech from PELCRA dataset, especially for *Whisper Large v1, v2, v3*. While *MMS* models performed well on BIGOS, they show increased WER for PELCRA, indicating challenges with spontaneous interactions. The efficiency of *Nemo Multilang* and *Nemo Conformer* is also notable, though their advantage is reduced for conversational speech.

Overall, larger models are particularly beneficial for handling the variability of conversational datasets like PELCRA compared to read speech.

### 4.1.4 Impact of speech rate on WER

Figures 5b and 6b show the relationship between Word Error Rate (WER) and speech rate for the BIGOS (read speech) and PELCRA (conversational speech) datasets. In both, WER decreases at moderate rates and rises for extreme speeds.

For BIGOS, WER is lowest between 1-2 words per second, with *assembly_best* slightly outperforming *whisper_large_v3*. PELCRA shows a broader range up to 8 words per second, with WER lowest between 2-4 words per second, but both models struggle at extreme rates, particularly *whisper_cloud*. Manual inspection revealed a stronger tendency for the *whisper* model to produce *hallucinations* in either short recordings with high speech rates or long recordings with slowly pronounced words. Overall, conversational speech presents higher WER due to variability, while moderate speech rates yield optimal performance in both datasets.

## 4.2 Implications

The developed data curation and evaluation system offers the following benefits for the research community:

- Establishes a standard for evaluating Polish ASR systems, enhancing reproducibility.
- Facilitates better use of datasets, promoting focused research. As of October 30th 2024, *BIGOS V2* dataset had over 6,500 downloads, while *PELCRA for BIGOS* had over 1,500 downloads.
- Encourages data sharing and collaboration, improving resources and progress.
- Identifies gaps, such as the need for detailed metadata and semantic metrics, guiding future studies.

Advantages for industry include:

- Informs public about strengths and weaknesses of available ASR system.
- Proposes a standard evaluation procedure to increase evaluation efficiency.
- Showcases the importance of normalization and utilization of metadata for analysis.
- Provides incentive to companies to showcase superior performance on a public benchmark for marketing purposes.

## 4.3 Limitations and challenges

The reliability of results may be affected if recordings from popular datasets, such as *Common Voice* and *MLS*, were included in training of the evaluated systems. To address this, new, non-public test recordings should be added to the benchmark dataset. Future research should also include manual transcriptions and annotations to ensure test data quality. Manual and automatic error classification and correction [40] can also be explored. Adding semantically informed metrics could offer additional insights into task-specific accuracy. [37, 35] Incorporating recordings that represent diverse usage conditions and Polish speaker demographics should improve reliability of assessing ASR systems robustness [14] and sociodemographic bias. [2, 1] Lastly, newly released systems and model updates could be systematically evaluated and compared with longitudinal studies in other languages [36].

## 5 Conclusion

The research addresses the issue of limited dataset usage for Polish benchmarking by offering a curated benchmark set derived from 24 publicly available datasets. The evaluation of 9 ASR systems and 25 models revealed notable performance differences between model sizes and speech types. This work improves reproducibility and directs future ASR advancements by providing public access to data catalogs, curated datasets, evaluation tools, and dashboards with comprehensive benchmarking results covering 8 scenarios. Specific methods and tools has potential to be reused for other low-resource languages.

## 6    Acknowledgments

The author gratefully acknowledges the original dataset creators for sharing their work openly and allowing its curation in this resource. BibTeX citations for the original authors are provided on curated datasets web pages, and the author hopes that users of datasets curated in *BIGOS* format will include references to the original sources. [20] The author also extends thanks to anonymous reviewers for their insightful feedback on the first version of this manuscript.

## 7    References

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

# 8 Appendices

Provide additional data, tools' documentation, and other supplementary materials that are relevant but not central to the article's narrative.


# A    Additional information required by organizers

In the Appendix, we provide additional information. This section will often be part of the supplemental material. Please see the call on the NeurIPS website for links to additional guides on dataset publication.

Submission introducing new datasets must include the following in the supplementary materials:

1. Dataset documentation and intended uses. Recommended documentation frameworks include datasheets for datasets, dataset nutrition labels, data statements for NLP, and accountability frameworks.

2. URL to website/platform where the dataset/benchmark can be viewed and downloaded by the reviewers.

3. URL to Croissant metadata record documenting the dataset/benchmark available for viewing and downloading by the reviewers. You can create your Croissant metadata using e.g. the Python library available here: https://github.com/mlcommons/croissant

4. Author statement that they bear all responsibility in case of violation of rights, etc., and confirmation of the data license.

5. Hosting, licensing, and maintenance plan. The choice of hosting platform is yours, as long as you ensure access to the data (possibly through a curated interface) and will provide the necessary maintenance.

# B    Additional information relevant to submitted article

## B.1    Polish ASR benchmarks overview

This section presents ASR benchmarks for the Polish language reported in the public domain as of March 2024:

- BOR (*BOR POLSL PS 18*) [25]
- PolEval 19 ASR challenge (PolEval PJATK 19) [15]
- DiaBiz commercial ASR systems benchmark [29]
- Medical PG [41]
- Medical PŚ [17]

The benchmarks described in this study, as well as those performed on a previous version of the BIGOS dataset [10], are not included.

Table 3 provides a summary of reported Polish ASR benchmarks, listing each benchmark by year, models evaluated, the best model, and observations on performance.

Table 3: Reported benchmarks for Polish ASR Systems as of March 2024

| Benchmark | Year | Models | Best Model | Lowest WER | Observations |
|---|---|---|---|---|---|
| BOR POLSL PS 18 | 2018 | ARM, Skrybot, Google | Google | clean-50% noisy-90% | Tested systems are not accurate enough for training government agents. |
| PolEval PJATK 19 | 2019 | GOLEM, ARM-1, SGMM2, tri2a, clarin-pl-studio, clarin-pl-sejm | GOLEM | 11.80% | All systems except ARM-1 are based on Kaldi; all but clarin-pl use GMM models. Fixed systems used in-domain data only. |
| DiaBiz CLARIN Voicelab 22 | 2022 | Azure, Google, Voicelab | Azure | 10.50% | Azure achieved the best results (10.51 WER for both channels), followed by Voicelab's ASR (11.51 WER). Google's Polish ASR performed worse on the DiaBiz dataset (20.84 WER). Azure outperformed others in 8 of 9 domains, while Voicelab was slightly better for telecommunications customer support dialogs. |
| SpokesBiz CLARIN 23 | 2023 | Whisper (large) | Whisper | 20% | Whisper accuracy varies from typical evaluations on CommonVoice and FLEURS datasets. Recording quality and vocabulary domain greatly affect WER (15.2% – 26%). |
| Medical UW SOVVA PS 23 | 2023 | Azure, Google, Techmo | Google | 14% | All three ASR systems showed over 86% accuracy, with only a 1.7% difference between best and worst results. |
| Medical PG 23 | 2023 | Azure, Google, Whisper (large-v2) | Azure | 56% | Models are unsuitable for medical records, case descriptions, or treatment prescriptions due to high error rates (WER 56%, CER 16%). |

Table 4: Overview of ASR Use Cases in Polish ASR Benchmarks

| Benchmark | Use Cases |
|---|---|
| BOR POLSL PS 18 | Voice Control |
| PolEval PJATK 19 | Oration |
| DiaBiz CLARIN Voicelab 22 | Conversations |
| SpokesBiz CLARIN 23 | Conversations, Meetings, Orations |
| Medical UW SOVVA PS 23 | Dictation |
| Medical PG 23 | Dictation |

Table 4 offers an overview of the specific use cases for these benchmarks, detailing primary applications across each evaluation.

Table 5 summarizes benchmarks from 2018 to 2023, including systems, datasets, and metrics utilized.

In Table 6, a breakdown is provided of domains, speech types, audio sources, and recording devices across benchmarks.

Table 5: Summary of Public Domain ASR Benchmarks (2018–2023)

| Benchmark | Year | Systems | Datasets | Metrics Auto. | Metrics Manual |
|---|---|---|---|---|---|
| BOR POLSL PS 18 | 2018 | 3 | 1 | 3 | 0 |
| PolEval PJATK 19 | 2019 | 6 | 1 | 1 | 0 |
| DiaBiz CLARIN Voicelab 22 | 2022 | 3 | 7 | 3 | 0 |
| Medical PG 23 | 2023 | 3 | 1 | 6 | 0 |
| Medical UW PS 23 | 2023 | 3 | 1 | 5 | 3 |
| SpokesBiz CLARIN 23 | 2023 | 1 | 8 | 3 | 0 |

Table 6: Overview of Domains, Speech Types, Audio Sources, and Recording Devices

| Benchmark | Domain | Speech Types | Audio Sources | Audio Devices |
|---|---|---|---|---|
| BOR POLSL PS 18 | Government | Read | Field Rec. | Lavalier Mic |
| PolEval PJATK 19 | Parliament | Read | Field Rec. | Venue Mic |
| DiaBiz CLARIN 22 | Cust. Support | Spontaneous | Phone calls | Phone |
| SpokesBiz CLARIN 23 | Various | Spontaneous | Podcasts | Various |
| Medical UW PS 23 | Medical | Read | Field Rec. | Lavalier Mic |
| Medical PG 23 | Medical | Read | Field Rec. | Lavalier Mic |

Table 7 reports dataset size, domain count, number of recordings, and speakers involved for each benchmark.

Table 7: Dataset Size, Number of Domains, Recordings, and Speakers

| Benchmark | Audio (hours) | Domains | Recordings | Speakers |
|---|---|---|---|---|
| BOR POLSL PS 18 | 1 | 1 | 140 | 18 |
| PolEval PJATK 19 | 1 | 1 | 29 | 29 |
| DiaBiz CLARIN 22 | 41 | 7 | 400 | 151 |
| SpokesBiz CLARIN 23 | 52 | 7 | 79 | 79 |
| Medical UW PS 23 | 1 | 1 | 1000 | No Info |
| Medical PG 23 | 1 | 1 | 1200 | 10 |

Table 8 highlights the acoustic conditions and speaker metadata available in these Polish ASR benchmarks.

Table 8: Acoustic Conditions, Annotations, and Speaker Meta-Data Across Polish ASR Benchmarks

| Benchmark | Acoustic Conditions | Speaker Meta-Data |
|---|---|---|
| BOR POLSL PS 18 | Mixed | None |
| PolEval PJATK 19 | Mixed | None |
| DiaBiz CLARIN 22 | Mixed | Age, gender, education |
| SpokesBiz CLARIN 23 | Mixed | Age, gender, education |
| Medical UW PS 23 | Clean | Age, gender, region |

Table 9 details the metrics used, both automated and human-evaluated, to assess system performance.

Table 10 presents benchmarks from 2018 to 2023 with the number and details of ASR systems evaluated, totaling 19 system-model combinations.

Table 11 shows the frequency of independent evaluations for ASR systems supporting Polish.

Table 12 lists ASR systems supporting Polish that, as of March 2024, lacked publicly reported evaluations. All systems, except *notta.ai* were included in this benchmark.

Finally, Table 13 categorizes system types, showing the range of commercial, public domain, and community-provided ASR systems benchmarked from 2018 to 2023.

Table 9: Overview of Metrics Employed in Polish ASR System Benchmarks

| Benchmark | Lexicon-Based Metrics | Annotation-Based Metrics |
|---|---|---|
| BOR POLSL PS 18 | SRR, WRR | None |
| PolEval PJATK 19 | WER | None |
| DiaBiz CLARIN Voicelab 22 | WER | None |
| SpokesBiz CLARIN 23 | WER, MER, WIL | None |
| Medical UW PS 23 | Accuracy, WER, LED, JWS | Error types |
| Medical PG 23 | WER, MER, WIL, CER, LED, Jaccard distance | None |

Table 10: Publicly Reported Evaluations of ASR Models for Polish Language

| Benchmark | Evaluated Systems | Models Evaluated |
|---|---|---|
| BOR POLSL PS 18 | ARM, Skrybot, Google | 3 |
| PolEval PJATK 19 | GOLEM, ARM-1, SGMM2, tri2a, clarin-pl-studio, clarin-pl-sejm | 6 |
| DiaBiz CLARIN Voicelab 22 | Azure, Google, Voicelab | 3 |
| SpokesBiz CLARIN 23 | Whisper (large) | 1 |
| Medical UW PS 23 | Azure, Google, Techmo | 3 |
| Medical PG 23 | Azure, Google, Whisper (large-v2) | 3 |
| **Total** | | **19** |

Table 11: Number of Reported Independent Evaluations and Benchmarks per System

| System | Benchmarks |
|---|---|
| azure_latest | 3 |
| google_default | 4 |
| skrybot_default | 1 |
| voicelab_default | 1 |
| arm_default | 2 |
| techmo_default | 1 |
| clarin_studio_kaldi_default | 1 |
| clarin_pl_sejm_default | 1 |
| golem_default | 1 |
| sgmm2_default | 1 |
| tri2a_default | 1 |
| whisper_local_large-v2 | 2 |
| **Total** | **19** |

# C Overview of tools for dataset management and ASR evaluation

## C.1 ASR speech datasets management tools

This section describes the most frequently used general or ASR-specific data management tools accessible under open licenses.

- **pandas** [20] is an open-source Python library that provides high-performance data manipulation and analysis tools. The objective of Pandas is to simplify the handling of data structures such as SQL tables, Excel, or text files, spanning from tabular data with different types of columns to time series and labeled matrices. The library gas two core data structures, the Series for one-dimensional data, and the DataFrame for two-dimensional data. *Pandas* excels in various data operations, such as managing missing data, modifying the size of data structures, aligning data based on labels, and grouping data for analysis. It also simplifies

Table 12: ASR Systems Supporting Polish Without Benchmark in the Public Domain as of March 2024

| System | Model | Type | License |
|---|---|---|---|
| google_v2 | long | Commercial | Proprietary |
| google_v2 | short | Commercial | Proprietary |
| google | latest_long | Commercial | Proprietary |
| google | latest_short | Commercial | Proprietary |
| google | command_and_search | Commercial | Proprietary |
| whisper_cloud | whisper-1 | Commercial | Proprietary |
| assembly_ai | best | Commercial | Proprietary |
| assembly_ai | nano | Commercial | Proprietary |
| notta.ai | default | Commercial | Proprietary |
| mms | 1b-all | Free | CC-BY-NC |
| mms | 1b-fl102 | Free | CC-BY-NC |
| mms | 1b-l1107 | Free | CC-BY-NC |
| nemo | stt_pl_fastconformer_hybrid_large_pc | Free | CC-BY |
| nemo | nemo_stt_multilingual_fastconformer... | Free | CC-BY |
| nemo | stt_pl_quartznet15x5 | Free | CC-BY |
| whisper_local | tiny | Free | MIT |
| whisper_local | base | Free | MIT |
| whisper_local | small | Free | MIT |
| whisper_local | medium | Free | MIT |
| whisper_local | large-v1 | Free | MIT |
| whisper_local | large-v3 | Free | MIT |
| wav2vec | xls-r-1b-polish | Free | Apache |
| wav2vec | large_xlsr-53-polish | Free | Apache |

Table 13: Types of ASR Systems Evaluated in Public Domain ASR Benchmarks (2018–2023)

| Benchmark | Year | System Types |
|---|---|---|
| BOR POLSL PS 18 | 2018 | Commercial |
| PolEval PJATK 19 | 2019 | Community Provided |
| DiaBiz CLARIN 22 | 2022 | Commercial |
| SpokesBiz CLARIN 23 | 2023 | Commercial |
| Medical UW PS 23 | 2023 | Commercial |
| Medical PG 23 | 2023 | Commercial + Public Domain |

the conversion of heterogeneous data forms into DataFrame objects, provides easy data slicing, indexing, concatenation, reshaping, and data fields renaming. Available under BSD 3-Clause license.

- **The Hugging Face datasets** [18] is a Python library designed to simplify data handling in ML projects. Its main benefit is the extensive support for public datasets in different formats and languages, which allows users to load the dataset with just one line of code. The library is also compatible with popular ML frameworks like *Numpy*, *Pandas*, *PyTorch*, *TensorFlow*, and *JAX*. *datasets* library facilitate efficient data preparation thanks to standardized data pre-processing tools that can handle datasets in various file formats. Furthermore, it simplifies the sharing of new datasets using the *HF datasets hub* [21]. Advanced library functionalities include:

  - handling large datasets beyond RAM capacity through memory-mapping,

  - smart caching to avoid redundant processing,

  - compatibility with different data types, including audio and image

  - streaming mode for efficient use of disk space and immediate data iteration.

---

[21]https://huggingface.co/datasets

- **Speech Data Explorer (SDE)** [5] is a tool for the exploration and analysis of speech datasets.[22] SDE was created by the NVIDIA team responsible for the development of the ASR system and the NLP framework Nemo.[23] Researchers used SDE to investigate errors and fine-tune the process of constructing a speech dataset using the *forced alignement* technique. The main features of SDE are:
    - calculating dataset statistics e.g., number of recordings, alphabet, vocabulary, duration-based histograms
    - dataset exploration with interactive data-tables for filtering and sorting
    - audio data inspection tools e.g., waveforms, spectrograms, audio playback
    - transcriptions and hypotheses analysis tools e.g., ASR accuracy metrics, alignments
    - audio signal measurements e.g., encoding, amplitude, spectrum

Summary information on tools for the management of ASR speech datasets is provided in Table 14.

Table 14: Tools for ASR Dataset Management

| Tool | Language | Features | License |
|---|---|---|---|
| Pandas | Python | Supports various data formats, data manipulation and analysis tools | BSD 3-Clause |
| Hugging Face Datasets | Python | Dataloaders for public datasets, large dataset handling, streaming | Apache 2.0 |
| Speech Data Explorer | Python | Dataset stats, audio inspection, transcription analysis, signal measurements | Apache 2.0 |

## C.2 ASR evaluation tools

This section outlines the most commonly used tools for the evaluation of ASR systems, which are available under permissive open-source licenses.

- **sclite**: Developed by the National Institute of Standards and Technologies (NIST), written in C, this tool uses the WER as its primary metric. Its features include speaker-level statistics, identification of commonly misrecognized words, and the ability to count hits, insertions, deletions, and substitutions. It also provides alignment capabilities. The software is available on *GitHub* and falls under NIST's software license.

- **jiwer**: A product of Jitsi, implemented in Python, JIWER calculates WER, along with Character Error Rate (CER), Match Error Rate (MER) and Word Information Lost (WIL) It supports aligning hypothesis and reference, as well as native support for text normalization transformations. The library is hosted on GitHub and released under the Apache 2.0 license.

- **asr-evaluation**: Created by Ben Lambert and also in Python, this tool measures WER, the word recognition rate WRR and the sentence error rate SER. It can handle simple normalization, removal of empty utterances, and calculation of the WER relative to the reference length. In addition, it generates confusion tables. Available on GitHub, *asr-evaluation* is licensed under Apache 2.0.

- **fstalign**: Developed by Rev and written in Python/C++, *fstalign* assesses WER and supports multiple input formats such as CTM, NLP, FST, and CSV. It natively supports text normalization and synonym handling and provides detailed error analysis based on metadata (WER tags) in NLP format. This tool is available on *GitHub* under the Apache 2.0 license.

- **evaluate**: From Hugging Face and built with Python/C++, this tool focuses on WER and is integrated with the Hugging Face *datasets* and *transformers*[24] libraries, enhancing its utility for users in the Hugging Face ecosystem. It can be found on *GitHub*, with an Apache 2.0 license.

---

[22]SDE User Guide

[23]NVIDIA NeMo ASR toolkit

[24]https://huggingface.co/docs/transformers/index

- **asr-evaluator** ASR evaluation tool from the NVIDIA's Nemo toolkit, with the following features:
  - On-the-fly data augmentation for ASR robustness evaluation.
  - Analysis of insertion, deletion, and substitution error rates.
  - Reliability assessment across metadata available, e.g. gender, audio length, etc.

Detail information on tools for the evaluation of ASR systems is provided in table 15.

Table 15: Tools for ASR Evaluation

| Tool | Author | Lang. | Metric(s) | Features | License |
|------|--------|-------|-----------|----------|---------|
| sclite | NIST | C | WER | Speaker-level stats, alignments, insertions, deletions | NIST |
| jiwer | Jitsi | Python | WER, CER, MER | Alignments, text normalization, CLI support | Apache 2.0 |
| asr-evaluation | B. Lambert | Python | WER, WRR, SER | WER by length, confusion tables | Apache 2.0 |
| fstalign | Rev | Python, C++ | WER | Supports CTM, NLP, CSV, native normalization | Apache 2.0 |
| evaluate | Hugging Face | Python | WER | Integrates with hfdatasets, transformers | Apache 2.0 |
| asr-evaluator | NVidia | Python | WER, CER | Nemo ASR models integration | Apache 2.0 |

## C.3 Evaluation metrics

ASR systems predictions were evaluated against target transcriptions using the following metrics:

- **Sentence Error Rate** (SER), which calculates the proportion of sentences that are not perfectly recognized, i.e., sentences that contain at least one error.

- **Word Error Rate** (WER), which is defined as the minimum number of operations (substitutions, insertions, and deletions) required to transform the system output into the reference transcript, divided by the total number of words in the reference. The result is expressed as a percentage. A lower WER indicates a more accurate system. The WER value can be greater than 100%.

- **Match Error Rate** (MER), which calculates the ratio of the total number of errors (substitutions, insertions, and deletions) to the total number of words in the reference and hypothesis (system output) transcripts. Unlike WER, which is normalized by the number of words in the reference, MER is normalized by the total number of words in both the reference and hypothesis. This makes the MER potentially less sensitive to the insertion of incorrect words by the ASR system, offering a different perspective on the accuracy of the system. MER value is equal to or less than 100%.

- **Character Error Rate** (CER), which calculates the minimum number of character-level operations (substitutions, insertions, and deletions) needed to change the system output to the reference transcript, divided by the total number of characters in the reference.

## C.4 Evaluated ASR system details

- **Google Cloud Speech-to-Text**[25] supports more than 125 languages and variants. Google's service offers several useful features, such as noise cancelation, support for streaming, automatic punctuation, and the capability to recognize specific phrases or words when provided with context (e.g., specialized vocabulary or formats for spoken numbers, addresses, years, currencies, etc.). For selected languages, it also provides domain-specific models, multichannel audio support, and filtering of profanity content. Two generations of service are available: v1[26] and v2.[27] For Polish, multiple model variants are available and were evaluated: *v1_default*, *v1_latest_long*, *v1_latest_short*, *v1_command_and_search*, *v2_long* and *v2_short*.

- **Microsoft's Azure Speech Service** [28] as of May 2023 supports more than 100 languages and variants. In addition to standard transcription, the Azure Speech Service supports continuous real-time speech recognition and provides robust noise reduction capabilities. It allows users to apply custom models to improve the accuracy of domain-specific terminology. Additional services include text search or analytics on transcribed content, as well as speaker diarization. The *latest default* model for Polish (dated for January 2023) was used, as no specialized model types support this language.

- **Whisper** [29] is an ASR system developed by the OpenAI company. It is trained on a large amount of weakly supervised multilingual and multitask data collected from the Internet. [32] According to the literature, Whisper is capable of handling different languages, dialects, and accents, demonstrating strong performance in diverse applications when evaluated on well-known benchmark datasets, e.g. Common Voice. [32] Whisper is available via a web API or as a pre-trained model for local use. Five versions of models of varying sizes are available for free download. The large model is available in 3 versions.

  source: For this benchmark, the commercial model available via API and eight locally run models were used.

- **NVIDIA NeMo** is the ASR system available as part of the Nemo toolkit[30]. Three models supporting the Polish language are available: *stt_pl_fastconformer_hybrid_large_pc*,

---

[25] https://cloud.google.com/speech-to-text

[26] https://cloud.google.com/speech-to-text/docs/speech-to-text-requests?hl=en

[27] https://cloud.google.com/speech-to-text/v2/docs?hl=en

[28] https://azure.microsoft.com/en-us/products/cognitive-services/speech-to-text

[29] https://github.com/openai/whisper/tree/main

[30] NEMO

Table 16: Model sizes and availability of English-only and Multilingual models.

| Size | Parameters | English-only model | Multilingual model |
|------|-----------|-------------------|-------------------|
| tiny | 39 M | Yes | Yes |
| base | 74 M | Yes | Yes |
| small | 244 M | Yes | Yes |
| medium | 769 M | Yes | Yes |
| large | 1550 M | No | Yes |

*stt_multilingual_fastconformer_hybrid_large_pc* and *stt_pl_quartznet15x5*. [16] Polish models were fine-tuned from English to Polish on the *Mozilla Common Voice (MCV)* dataset. [4] The authors report on 14 % WER on the *dev set* from the Polish MCV dataset. All models are available for free use under a CC-BY-NC license.

- **MMS**: Facebook AI's massive multilingual pre-trained model for speech ("MMS"). It was pre-trained on about 500,000 hours of speech data in more than 1,400 languages. [30] The MMS system supports over 1000 languages and other speech processing tasks such as *Text-to-Speech (TTS)* generation and *Speech Language Identification (LID)* [31]. The MMS system is available for free[32] under the CC-BY-NC 4.0 license. The following versions of the fine-tuned model of ASR are available:

  - *1b-fl102* - 1 billion parameter model fine-tuned on *FLEURS* Dataset [6]
  - *1b-l1107* - 1 billion parameter model fine-tuned *MMS-lab* Dataset. [30]
  - *1b-all* - 1 billion parameter model fine-tuned on *MMS-lab, FLEURS, CommonVoice, MLS* and *VoxPopuli* datasets. [4, 30, 31, 39]

- **Wav2Vec** is the automated speech recognition (ASR) system created by Facebook AI. It employs self-supervision to learn from unlabeled training data. Upon its launch in 2020, wav2vec2 exceeded the top semi-supervised approach with only a fraction of labeled training data. [9] Two models fine-tuned for Polish are available on the Hugging Face platform: *xls-r-1b-polish*[33] and *large_xlsr-53-polish*.

- **Assembly AI**[34] provides an advanced automatic speech recognition service supporting multiple languages. Key features include real-time transcription, automatic punctuation, and robust noise cancellation. The service supports domain-specific vocabulary through custom models, filtering of sensitive content and integration with various platforms via a web API. The system is designed to handle diverse accents and dialects, ensuring high accuracy across different use cases. According to the authors, their system "leverages a diverse training Dataset comprising unsupervised (12.5M hours), supervised (188k hours), and pseudo-labeled (1.6M hours) data across four languages". [33] It is also reported that the *Universal-1* model achieves comparative WER scores to larger and more computationally expensive models, such as Whisper large and Canary-1B. [33]. The amount of training data for Polish is not reported.

- **Open Whisper-style Speech Models**[35] (OWSM, pronounced as "awesome") are a series of speech foundation models developed by WAVLab at Carnegie Mellon University. They reproduce Whisper-style training using publicly available data and our open-source toolkit ESPnet.[36] The authors released data preparation scripts, training and inference code, pre-trained model weights and training logs in order to promote transparency and open science in large-scale speech pre-training.

---

[31] https://huggingface.co/spaces/mms-meta/MMS
[32] https://huggingface.co/facebook/mms-1b-all
[33] wav2vec2 fine-tuned to Polish
[34] Assembly AI
[35] OWSM homepage
[36] ESPnet

Table 17: ASR systems evaluated in the study.

| Shortname | System | Model |
|---|---|---|
| assembly_best | assembly_ai | best |
| assembly_nano | assembly_ai | nano |
| azure_latest | azure | latest |
| google_cmd_search | google | command_and_search |
| google_default | google | default |
| google_long | google | latest_long |
| google_short | google | latest_short |
| google_v2_long | google_v2 | long |
| google_v2_short | google_v2 | short |
| mms_all | mms | 1b-all |
| mms_102 | mms | 1b-fl102 |
| mms_1107 | mms | 1b-l1107 |
| nemo_multilang | nemo | stt_multilingual_fastconformer_hybrid_large_pc |
| nemo_pl_confromer | nemo | stt_pl_fastconformer_hybrid_large_pc |
| nemo_pl_quartznet | nemo | stt_pl_quartznet15x5 |
| w2v-53-pl | wav2vec2 | large-xlsr-53-polish |
| w2v-1b-pl | wav2vec2 | xls-r-1b-polish |
| whisper_cloud | whisper_cloud | whisper-1 |
| whisper_base | whisper_local | base |
| whisper_large_v1 | whisper_local | large-v1 |
| whisper_large_v2 | whisper_local | large-v2 |
| whisper_large_v3 | whisper_local | large-v3 |
| whisper_medium | whisper_local | medium |
| whisper_small | whisper_local | small |
| whisper_tiny | whisper_local | tiny |

## C.5 Normalization methods

Table 19 contains overview of scope of normalization of each available method.

## C.6 Evaluation results - PELCRA

Table 20 shows the impact of specific normalization methods for reduction of error rates for PELCRA dataset. Figure 7 shows a box plot of WER scores for PELCRA dataset.

Figure 8 shows a difference in WER for female and male recordings from PELCRA dataset. Figure 9 shows a standard deviation of WER for recordings originating from speakers in various age groups PELCRA dataset.

## C.7 Dataset splits details

Tables 22 and 23 present logic of data splits applied during curation for BIGOS and PELCRA datasets, respectively.

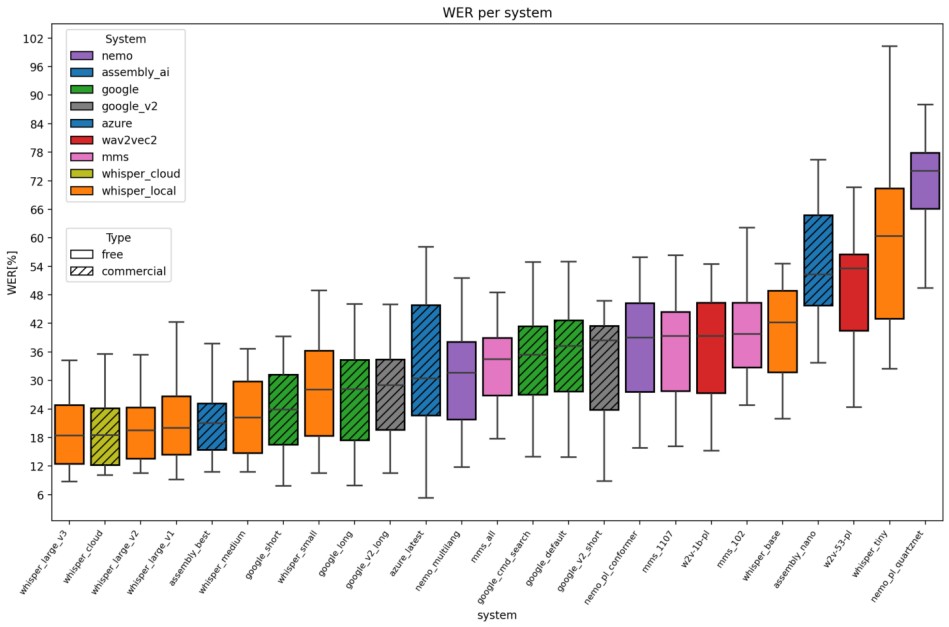

Figure 7: Box plot of WER for systems evaluated on the *PELCRA for BIGOS* dataset.

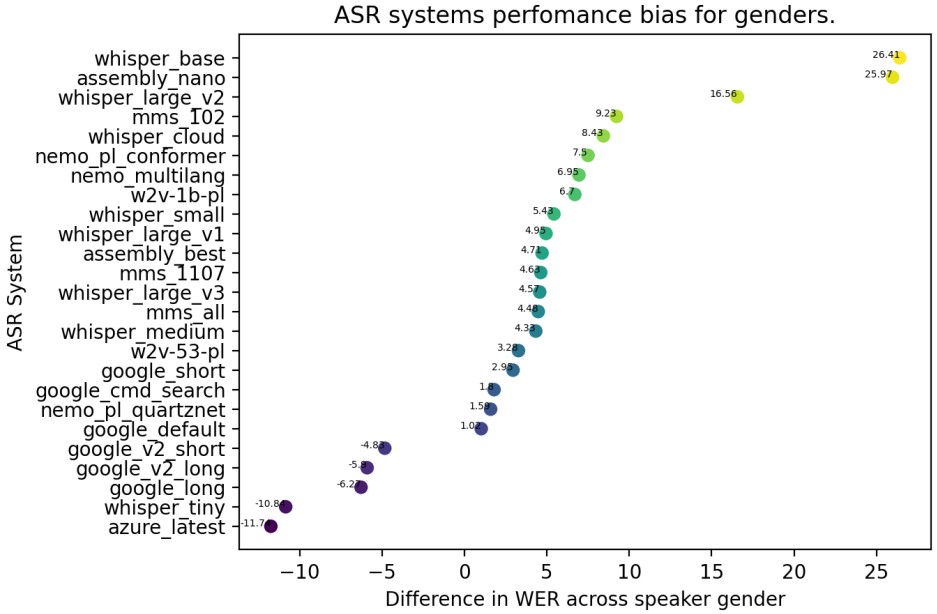

Figure 8: Difference in WER across speaker gender - PELCRA dataset.

Table 18: Evaluated ASR systems usage cost and license type.

| Shortname | Usage cost | License |
|---|---|---|
| assembly_best | commercial | Proprietary |
| assembly_nano | commercial | Proprietary |
| azure_latest | commercial | Proprietary |
| google_cmd_search | commercial | Proprietary |
| google_default | commercial | Proprietary |
| google_long | commercial | Proprietary |
| google_short | commercial | Proprietary |
| google_v2_long | commercial | Proprietary |
| google_v2_short | commercial | Proprietary |
| mms_all | free | CC-BY-NC |
| mms_102 | free | CC-BY-NC |
| mms_1107 | free | CC-BY-NC |
| nemo_multilang | free | CC-BY |
| nemo_pl_confromer | free | CC-BY |
| nemo_pl_quartznet | free | CC-BY |
| w2v-53-pl | free | Apache |
| w2v-1b-pl | free | Apache |
| whisper_cloud | commercial | Proprietary |
| whisper_base | free | MIT |
| whisper_large_v1 | free | MIT |
| whisper_large_v2 | free | MIT |
| whisper_large_v3 | free | MIT |
| whisper_medium | free | MIT |
| whisper_small | free | MIT |
| whisper_tiny | free | MIT |

Table 19: Methods of normalizing references and hypotheses.

| Normalization method | Scope |
|---|---|
| blanks removal | Elimination of superfluous white spaces. |
| lowercasing | Conversion of all characters to lowercase. |
| punctuation removal | Removal of punctuation symbols. |
| lexicon-based normalization | Removal of specific words e.g. fillers "um", "mhm" etc. Unification of spelling e.g. Kissindżer -> Kissinger |
| tags removal | Removal of tags e.g. 'trunc' in PELCRA dataset. |

## C.8  Dataset splits details

Table 24 presents metadata fields associated with each individual data item.

Table 20: Reduction of error rates caused by normalization of references and hypothesis for PELCRA dataset.

| Method | SER [p.p.] | WER [p.p.] | MER [p.p.] | CER [p.p.] | Average [p.p.] |
|---|---|---|---|---|---|
| blanks | -0.36 | 0 | 0 | 0 | -0.09 |
| tags | -0.37 | -0.16 | -0.19 | -0.13 | -0.21 |
| lowercase | -0.38 | -3.71 | -3.78 | -0.93 | -2.2 |
| punctation | -0.3 | -8.2 | -8.31 | -3.41 | -5.06 |
| dictionary | -1.52 | -2.4 | -2.23 | -2.17 | -2.08 |
| all | -9.15 | -15.65 | -15.78 | -6.25 | -11.71 |

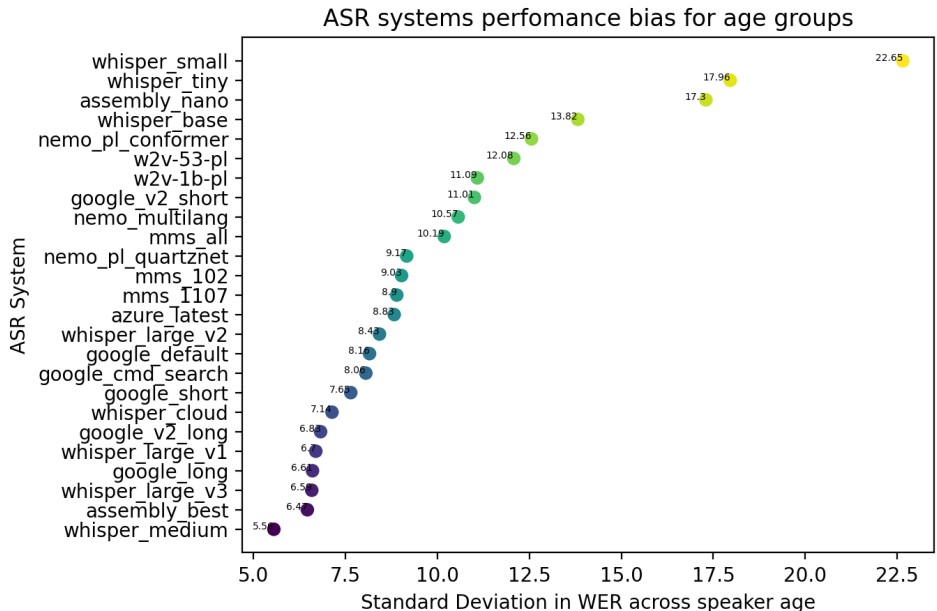

Figure 9: Difference in WER across speaker age - PELCRA dataset.

## C.9 Dataset contents details

Tables 25 and 26 present information on licensing and language coverage for BIGOS and PELCRA datasets, respectively.

## C.10 Dataset contents details

Tables 27 and 28 present information on domains, speech, and interaction types for BIGOS and PELCRA datasets, respectively.

## C.11 Dataset contents details

Tables 29 and 30 present information on sources, acoustic environments and audio recording devices for BIGOS and PELCRA datasets, respectively.

## C.12 Audio content size metrics

Tables 31 and 32 present information about number of available transcribed speech material, audio files and recorded speakers for BIGOS and PELCRA datasets, respectively.

| System | Twenties | Thirties | Fourties | Sixties | Seventies | Std. Dev |
|---|---|---|---|---|---|---|
| whisper_medium | 30.70 | 27.90 | 34.32 | 19.63 | 31.02 | 5.56 |
| assembly_best | 28.65 | 31.67 | 35.10 | 20.42 | 36.83 | 6.47 |
| whisper_large_v3 | 28.54 | 33.88 | 35.09 | 19.23 | 33.85 | 6.59 |
| google_long | 35.84 | 34.30 | 40.10 | 23.37 | 28.11 | 6.61 |
| whisper_large_v1 | 30.60 | 33.39 | 38.57 | 20.62 | 27.44 | 6.70 |
| google_v2_long | 36.76 | 36.98 | 41.03 | 23.68 | 30.18 | 6.83 |
| whisper_cloud | 24.42 | 32.38 | 36.42 | 18.71 | 32.26 | 7.14 |
| google_short | 30.14 | 30.96 | 46.12 | 25.99 | 32.33 | 7.65 |
| google_cmd_search | 45.27 | 46.16 | 55.02 | 32.88 | 41.07 | 8.06 |
| google_default | 47.15 | 47.50 | 55.18 | 34.47 | 38.51 | 8.16 |
| whisper_large_v2 | 24.79 | 37.72 | 33.14 | 15.70 | 29.40 | 8.43 |
| azure_latest | 42.68 | 38.16 | 41.03 | 23.73 | 25.97 | 8.83 |
| mms_1107 | 47.43 | 51.72 | 61.20 | 37.36 | 54.86 | 8.90 |
| mms_102 | 47.48 | 57.60 | 64.74 | 41.31 | 52.47 | 9.03 |
| nemo_pl_quartznet | 75.10 | 74.76 | 83.85 | 58.77 | 69.89 | 9.17 |
| mms_all | 48.30 | 45.20 | 58.40 | 30.17 | 48.33 | 10.19 |
| nemo_multilang | 38.60 | 43.66 | 53.57 | 24.45 | 42.65 | 10.57 |
| google_v2_short | 41.12 | 41.96 | 53.69 | 26.48 | 28.98 | 11.01 |
| w2v-1b-pl | 40.59 | 59.48 | 61.57 | 36.66 | 51.29 | 11.09 |
| w2v-53-pl | 61.85 | 62.14 | 71.46 | 40.05 | 50.87 | 12.08 |
| nemo_pl_conformer | 45.96 | 52.00 | 64.78 | 30.02 | 50.91 | 12.56 |
| whisper_base | 54.67 | 48.19 | 71.40 | 35.04 | 62.24 | 13.82 |
| assembly_nano | 62.85 | 60.88 | 78.18 | 56.22 | 98.55 | 17.30 |
| whisper_tiny | 88.47 | 88.03 | 80.52 | 45.12 | 70.87 | 17.96 |
| whisper_small | 40.68 | 32.96 | 83.03 | 24.65 | 39.38 | 22.65 |

Table 21: WER across age groups - *PELCRA for BIGOS* dataset.

Table 22: Metadata and partitioning of source datasets - *BIGOS V2 dataset*

| Subset | Original partitioning | BIGOS split process | Entity for BIGOS split |
|---|---|---|---|
| google-fleurs-22 | train, test, dev | original splits preserved | N/A |
| polyai-minds14-21 | none | pseudorandom | audio file id |
| pjatk-clarin_mobile-15 | none | pseudorandom | session (speaker id) |
| pjatk-clarin_studio-15 | none | pseudorandom | session (speaker id) |
| pwr-azon_read-20 | none | pseudorandom | session (speaker id) |
| pwr-azon_spont-20 | none | pseudorandom | session (speaker id) |
| fair-mls-20 | train, test, dev | original splits preserved | N/A |
| mozilla-cv15-23 | train, test, dev | original splits preserved | N/A |
| mailabs-corpus_librivox-19 | none | pseudorandom | audio file id |
| pwr-maleset-unk | none | pseudorandom | audio file id |
| pwr-shortwords-unk | none | pseudorandom | audio file id |
| pwr-viu-unk | none | pseudorandom | audio file id |

Table 23: Metadata and partitioning of source datasets - *PELCRA for BIGOS* dataset

| Subset | Original partitioning | BIGOS split process | Entity for BIGOS split |
|---|---|---|---|
| ul-diabiz_poleval-22 | train, test, dev | original splits preserved | N/A |
| ul-spokes_biz_bio-23 | none | pseudorandom | recording id |
| ul-spokes_biz_int-23 | none | pseudorandom | recording id |
| ul-spokes_biz_luz-23 | none | pseudorandom | recording id |
| ul-spokes_biz_pod-23 | none | pseudorandom | recording id |
| ul-spokes_biz_pres-23 | none | pseudorandom | recording id |
| ul-spokes_biz_vc-23 | none | pseudorandom | recording id |
| ul-spokes_biz_vc2-23 | none | pseudorandom | recording id |
| ul-spokes_biz_wyw-23 | none | pseudorandom | recording id |
| ul-spokes_mix_emo-18 | none | pseudorandom | recording id |
| ul-spokes_mix_luz-18 | none | pseudorandom | recording id |
| ul-spokes_mix_parl-18 | none | pseudorandom | recording id |

Table 24: Attributes in the BIGOS utterance data object

| Field name | Description |
|---|---|
| audioname | Standardized unique identifier for each audio recording in the dataset. |
| split | Indicates the dataset split the recording belongs to: train, test or validation. |
| dataset | Source dataset identifier. |
| ref_orig | The original transcript associated with the audio recording. |
| ref_spoken | Transcription in the spoken domain format. |
| ref_written | Transcription in the written domain format. |
| audio | Object for storing audio data in HF datasets format. |
| sampling_rate | The sampling rate of the audio recording in the dataset. Can be the same as the original or adjusted for standardization. |
| samplingrate_orig | The original sampling rate of the audio recording. |
| speaker_id | A unique identifier of the speaker in the recording. |
| audiopath_bigos | The relative path to the audio file from distributed data archive. |
| audiopath_local | The absolute path to the extracted audio file, typically in the default hf datasets cache directory. |
| audio_duration_samples | Recording duration in samples. |
| audio_duration_seconds | Recording duration in seconds. |
| speaker_gender | Information about the speaker's gender in the CommonVoice format. |
| speaker_age | Information about the speaker's age in CommonVoice format. |
| speech_rate_words | Speech rate expressed in words per seconds. |
| speech_rate_chars | Speech rate expressed in characters per seconds. |
| utterance_length_words | Length of the utterance in words. |
| utterance_length_chars | Length of the utterance in characters. |

Table 25: BIGOS V2 dataset subset license and language coverage.

| Dataset | Codename | License | Languages |
|---|---|---|---|
| Clarin Studio | pjatk-clarin_studio-15 | CC-BY | monolingual |
| Clarin Mobile | pjatk-clarin_mobile-15 | CC-BY | monolingual |
| Munich AI Labs LibriVox | mailabs-corpus_librivox-19 | Proprietary | multilingual |
| Mozilla Common Voice | mozilla-common_voice_15-23 | CC-0 | multilingual |
| Multilingual Librispeech | fair-mls-20 | CC-BY | multilingual |
| Azon Read | pwr-azon_read-20 | CC-BY-SA | monolingual |
| Azon Spontaneous | pwr-azon_spont-20 | CC-BY-SA | monolingual |
| PWR Male Set | pwr-maleset-unk | Public domain | monolingual |
| PWR Short Words | pwr-shortwords-unk | Public domain | monolingual |
| PWR Very Important Utterances | pwr-viu-unk | Public domain | monolingual |
| Google FLEURS | google-fleurs-22 | CC-BY | multilingual |
| PolyAI Minds14 | polyai-minds14-21 | CC-BY | multilingual |

Table 26: PELCRA for BIGOS dataset subset license and language coverage.

| Dataset | Codename | License | Languages |
|---|---|---|---|
| DiaBiz ASR PolEval 22 | ul-diabiz_poleval-22 | Public domain | monolingual |
| SpokesBiz CBIZ_BIO | ul-spokes_biz_bio-23 | CC-BY-NC-ND | monolingual |
| SpokesBiz CBIZ_INT | ul-spokes_biz_int-23 | CC-BY-NC-ND | monolingual |
| SpokesBiz CBIZ_LUZ | ul-spokes_biz_luz-23 | CC-BY-NC-ND | monolingual |
| SpokesBiz CBIZ_POD | ul-spokes_biz_pod-23 | CC-BY-NC-ND | monolingual |
| SpokesBiz CBIZ_PRES | ul-spokes_biz_pres-23 | CC-BY-NC-ND | monolingual |
| SpokesBiz CBIZ_VC | ul-spokes_biz_vc-23 | CC-BY-NC-ND | monolingual |
| SpokesBiz CBIZ_VC2 | ul-spokes_biz_vc2-23 | CC-BY-NC-ND | monolingual |
| SpokesBiz CBIZ_WYW | ul-spokes_biz_wyw-23 | CC-BY-NC-ND | monolingual |
| SpokesMix PELCRA_EMO | ul-spokes_mix_emo-18 | CC-BY | monolingual |
| SpokesMix PELCRA_LUZ | ul-spokes_mix_luz-18 | CC-BY | monolingual |
| SpokesMix PELCRA_PARL | ul-spokes_mix_parl-18 | CC-BY | monolingual |

Table 27: BIGOS V2 dataset subset domains and speech types.

| Codename | Domain | Speech type | Interaction type |
|---|---|---|---|
| pjatk-clarin_studio-15 | open domain | read | monolog |
| pjatk-clarin_mobile-15 | open domain | read | monolog |
| mailabs-corpus_librivox-19 | audiobook | read | monolog |
| mozilla-common_voice_15-23 | open domain | read | monolog |
| fair-mls-20 | audiobook | read | monolog |
| pwr-azon_read-20 | scientific | read | monolog |
| pwr-azon_spont-20 | scientific | spontaneous | monolog |
| pwr-maleset-unk | commands | read | monolog |
| pwr-shortwords-unk | commands | read | monolog |
| pwr-viu-unk | commands | read | monolog |
| google-fleurs-22 | wikipedia | read | monolog |
| polyai-minds14-21 | banking | read | monolog |

Table 28: PELCRA for BIGOS dataset subset domains and speech types.

| Codename | Domain | Speech type | Interaction type |
|---|---|---|---|
| ul-diabiz_poleval-22 | customer service | spontaneous | dialog |
| ul-spokes_biz_bio-23 | open domain | spontaneous | dialog |
| ul-spokes_biz_int-23 | open domain | spontaneous | dialog |
| ul-spokes_biz_luz-23 | open domain | spontaneous | dialog |
| ul-spokes_biz_pod-23 | open domain | spontaneous | dialog |
| ul-spokes_biz_pres-23 | open domain | spontaneous | dialog |
| ul-spokes_biz_vc-23 | open domain | spontaneous | dialog |
| ul-spokes_biz_vc2-23 | open domain | spontaneous | dialog |
| ul-spokes_biz_wyw-23 | open domain | spontaneous | dialog |
| ul-spokes_mix_emo-18 | open domain | spontaneous | dialog |
| ul-spokes_mix_luz-18 | open domain | spontaneous | dialog |
| ul-spokes_mix_parl-18 | open domain | spontaneous | monolog |

Table 29: BIGOS V2 dataset subset speakers, environments, and devices.

| Codename | Speech source | Acoustic environment | Audio device |
|---|---|---|---|
| pjatk-clarin_studio-15 | volunteers | quiet | studio mic |
| pjatk-clarin_mobile-15 | volunteers | quiet | mobile phone |
| mailabs-corpus_librivox-19 | volunteers | quiet | various |
| mozilla-common_voice_15-23 | crowd | various | various |
| fair-mls-20 | volunteers | various | various |
| pwr-azon_read-20 | volunteers | quiet | studio mic |
| pwr-azon_spont-20 | public speakers | mixed | lavalier |
| pwr-maleset-unk | volunteers | quiet | studio mic |
| pwr-shortwords-unk | volunteers | quiet | studio mic |
| pwr-viu-unk | volunteers | quiet | studio mic |
| google-fleurs-22 | volunteers | quiet | mobile phone |
| polyai-minds14-21 | crowd | quiet | mobile phone |

Table 30: PELCRA for BIGOS subsets speakers, environments, and devices.

| Codename | Speech source | Acoustic environment | Audio device |
|---|---|---|---|
| ul-diabiz_poleval-22 | volunteers | quiet | telephone |
| ul-spokes_biz_bio-23 | volunteers | quiet | lavalier mic |
| ul-spokes_biz_int-23 | volunteers | quiet | lavalier mic |
| ul-spokes_biz_luz-23 | volunteers | quiet | lavalier mic |
| ul-spokes_biz_pod-23 | public speakers | quiet | various |
| ul-spokes_biz_pres-23 | public speakers | quiet | various |
| ul-spokes_biz_vc-23 | volunteers | quiet | lavalier mic |
| ul-spokes_biz_vc2-23 | volunteers | quiet | lavalier mic |
| ul-spokes_biz_wyw-23 | volunteers | quiet | lavalier mic |
| ul-spokes_mix_emo-18 | volunteers | quiet | lavalier mic |
| ul-spokes_mix_luz-18 | volunteers | quiet | lavalier mic |

Table 31: Audio content size metrics for *BIGOS V2 dataset*

| Subset | Size [hours] | Samples | Speakers |
|---|---|---|---|
| fair-mls-20 | 107.86 | 26072 | 24 |
| google-fleurs-22 | 12.07 | 3937 | 3 |
| mailabs-corpus_librivox-19 | 32.14 | 14862 | 2 |
| mozilla-common_voice_15-23 | 53.00 | 36910 | 2920 |
| pjatk-clarin_mobile-15 | 12.48 | 3495 | 117 |
| pjatk-clarin_studio-15 | 56.43 | 13810 | 553 |
| polyai-minds14-21 | 3.07 | 562 | 3 |
| pwr-azon_read-20 | 5.72 | 2788 | 29 |
| pwr-azon_spont-20 | 2.14 | 456 | 27 |
| pwr-maleset-unk | 6.38 | 4738 | 3 |
| pwr-shortwords-unk | 1.43 | 939 | 3 |
| pwr-viu-unk | 1.04 | 2703 | 3 |
| total | 293.76 | 111272 | 3945 |

Table 32: Audio content size metrics for *PELCRA for BIGOS* dataset

| Subset | Size [hours] | Samples | Speakers |
|---|---|---|---|
| ul-diabiz_poleval-22 | 9.83 | 8950 | 170 |
| ul-spokes_biz_bio-23 | 137.98 | 54917 | 158 |
| ul-spokes_biz_int-23 | 2.25 | 1109 | 9 |
| ul-spokes_biz_luz-23 | 74.27 | 41966 | 158 |
| ul-spokes_biz_pod-23 | 55.00 | 22807 | 113 |
| ul-spokes_biz_pres-23 | 32.25 | 17174 | 55 |
| ul-spokes_biz_vc-23 | 52.07 | 45272 | 78 |
| ul-spokes_biz_vc2-23 | 81.04 | 25802 | 84 |
| ul-spokes_biz_wyw-23 | 28.21 | 11357 | 38 |
| ul-spokes_mix_emo-18 | 25.61 | 24329 | 40 |
| ul-spokes_mix_luz-18 | 18.74 | 20919 | 21 |
| ul-spokes_mix_parl-18 | 12.27 | 8656 | 48 |
| total | 529.52 | 283258 | 972 |

