# OpenReview forum: "BIGOS V2 Benchmark for Polish ASR: Curated Datasets and Tools for Reproducible Evaluation"
_NeurIPS.cc/2024/Datasets_and_Benchmarks_Track — NeurIPS 2024 Track Datasets and Benchmarks Poster_

### Official Review · Reviewer_B4Cv · 2024-07-03
**Dear Authors**

**Rating:** 5
**Confidence:** 5
**Correctness:** Not Sure
**Clarity:** Not Sure

**Review:**

The paper offers a detailed framework for surveying, cataloging, and curating speech datasets, which enhances the reproducibility of ASR system evaluations.

While the framework and its application are well-documented, the paper lacks significant academic insights or novel contributions.  The claims made are not supported by diverse experiments or substantial evidence, reducing the paper’s novelty. Much of the content reiterates known industry practices without offering new perspectives or substantial academic contributions. To enhance its academic value, the paper should incorporate comprehensive experiments and provide more rigorous evidence to support its claims.

**Strengths:**

See Review

**Additional Feedback:**

N/A

**Documentation:**

Not Sure

**Ethics:**

Not Sure

**Limitations:**

Not Sure

**Opportunities For Improvement:**

[1] The current state of the paper presents a series of experimental results without delving into deeper analysis. It would be beneficial to include detailed probing experiments that explore the dataset's strengths and limitations more thoroughly. This could involve comparative analyses with existing datasets to highlight unique features or potential improvements the new dataset offers. Although this paper is not about proposing or constructing a new dataset, I believe that a clear indication of new directions is necessary.

[2] The paper would benefit from integrating more theoretical discussions that link the dataset characteristics with potential impacts on the ASR community. This could include hypotheses on how the dataset might influence future research directions or practical applications, supported by experimental data.

**Relation To Prior Work:**

Not Sure

**Summary And Contributions:**

This paper presents a comprehensive framework for curating speech datasets and evaluating Automatic Speech Recognition (ASR) systems, focusing on the Polish language. It curates over 24 datasets and evaluates 25 ASR system-model combinations, providing extensive performance comparisons and insights.

---

> ### Comment · Reviewer_B4Cv · 2024-08-23
> **Dear Authors**
>
> I realize that there were aspects of this track that I misunderstood. As a result, I will be raising my score.

---

> > ### Author Response · Authors · 2024-08-29
> > **Thank you for feedback and revised score**
> >
> > Dear Reviewer,
> >
> > Thank you for your thoughtful feedback. Here are my responses to your comments:
> >
> > **Academic Insights and Novel Contributions:**
> > The paper introduces a comprehensive framework for curating speech datasets and evaluating ASR systems, specifically targeting the Polish language. While the focus is on curating and benchmarking rather than developing new datasets from scratch, the aim is to provide a systematic, reproducible approach that advances data-centric AI methods and enhances dataset utility for the ASR community.
> >
> > ### **Alignment with Track Requirements:**
> >
> > The presented work was designed to meet several of the track's key requirements, as outlined below:
> >
> > - **New datasets, or carefully and thoughtfully designed (collections of) datasets based on previously available data:**
> >   The work introduces a new collection based on previously available data, resulting in the largest-to-date unified and easily accessible speech dataset for Polish ASR development.
> >
> > - **Data-centric AI methods and tools, e.g., to measure and improve data quality or utility, or studies in data-centric AI that bring important new insights:**
> >   The dataset's utility has been enhanced through careful curation and redistribution via a popular open data platform, resulting in improved accessibility and broader adoption.
> >
> > - **Advanced practices in data collection and curation that are of general interest, even if the data itself cannot be shared:**
> >   The paper outlines a detailed framework for dataset curation, adhering to advanced practices in data collection. These practices are applicable to other languages and domains, thereby providing value even when specific data cannot be shared.
> >
> > - **Frameworks for responsible dataset development, audits of existing datasets, identifying significant problems with existing datasets and their use:**
> >   The work catalogs and analyzes available datasets based on documentation (as seen in related work: the [Polish Speech datasets survey](https://huggingface.co/spaces/amu-cai/pl-asr-survey), [article](https://www.degruyter.com/document/doi/10.1515/psicl-2023-0019/html)) and through content inspection (referenced in the [BIGOS dataset cards](https://huggingface.co/spaces/amu-cai/amu-bigos-data-dash)).
> >
> > - **Benchmarks on new or existing datasets, as well as benchmarking tools:**
> >   The work provides both datasets and tools for benchmarking, which are openly available with comprehensive documentation to enable replication by other researchers.
> >
> > - **In-depth analyses of machine learning challenges and competitions (by organizers and/or participants) that yield important new insights:**
> >   The paper includes a systematic analysis of existing ASR systems on a novel dataset, providing significant insights. It offers the most comprehensive evaluation to date of Polish ASR systems, covering both commercial and open-source models.
> >
> > ### **Opportunities for Improvement:**
> >
> > 1. **Detailed Experiments and Analysis:**
> >    In future work, probing experiments to explore dataset strengths and limitations will be considered. Comparative analyses with existing commercial datasets and the identified gaps of open datasets were provided in the related work (the [Polish Speech datasets survey](https://huggingface.co/spaces/amu-cai/pl-asr-survey), [article](https://www.degruyter.com/document/doi/10.1515/psicl-2023-0019/html)). The final revision will emphasize comparison with similar work (ASR benchmarking on open data) for Polish performed so far.  Future work will also include comparisons with similar efforts for other CEE languages.
> >
> > 2. **Theoretical Discussions and Implications:**
> >   The manuscript will be expanded to include discussions about current and potential impacts, such as evidence of community adoption (e.g., number of downloads, citations, and contributions from researchers developing Polish ASR systems). Depending on feasibility, the revised manuscript or future work will discuss community contributions (e.g., newly contributed datasets, reported systems to include in the benchmark, or new evaluation scenarios like noise mixing and audio perturbation), demonstrating the evolving nature of this benchmark.
> >
> > Thank you again for your valuable feedback.
> >
> > Best regards,
> > MJ

---

### Official Review · Reviewer_j19w · 2024-07-22
**Valuable work for expanding the reach of AI; paper text could be improved**

**Rating:** 7
**Confidence:** 3
**Correctness:** Yes.
**Clarity:** Yes, with some gaps as highlighted.

**Review:**

Quality: This work is high-quality and its clear the author has put time and effort into evaluating existing datasets and putting together meaningful criteria to prepare new ones that makes it easier for the community to develop new models.

Clarity is lacking in some instances; these will be highlighted in the sections below.

Originality

-- This is very unclear, in particular to the author's previous works, some of which aren't even included in the citations:
https://annals-csis.org/proceedings/2023/drp/pdf/1609.pdf

The BIGOS dataset is mentioned as a contribution of the current paper, whereas it was already claimed as a contribution of an earlier paper above.

-- There is no comparison to how this work compares to others developed for similar languages and how the scope differs in this work versus others. Thus it is hard for the reader to have a calibrated understanding of this work as compared to others in the area.

Significance: Overall a valuable work and in the right direction, but may need further work to improve this paper as a standalone contribution.

**Strengths:**

Strengths mentioned and highlighted above.

**Additional Feedback:**

Solid work, but improvements required for publication.

**Documentation:**

For this Github:
https://github.com/goodmike31/pl-asr-speech-data-survey

There is no license and no maintenance plan is mentioned.

**Ethics:**

No. It looks like the author has reviewed the licenses under which the existing datasets are available.

**Limitations:**

Yes.

**Opportunities For Improvement:**

Overall the model benchmarking sections should be described in further details. If code could be released where the training results could be reproduced, that would be the ideal scenario. On the existing text:

-- Section 2.4.2: The metrics SER, WER, ... should be defined and some context should be provided on the meaning of each metric.

-- Section 2.4.2: There should be references for the models Google STT, Azure STT, Whisper, etc. It's very strange that citations are missing here.

-- Figures 3 and 4: These are completely illegible. Furthermore, the process of generating these plots is not really described. Again, if the code that produces these plots could be released, that would significantly increase the value of this benchmark. As a minimum training hyperparameters should be mentioned in the main body.

-- Section 3.4: Some descriptions of the results are given here, but no analysis as to the why the results turned out like so.

**Relation To Prior Work:**

Major gaps have been identified.

**Summary And Contributions:**

This work is focused on giving developers / researchers the necessary tools to build ASR models for the Polish language. The overall premise of the work is very important, as beyond a handful of languages, it is incredibly difficult to build high-quality AI models for other languages due to lack of top-notch datasets. The contributions of this paper include:

-- Two datasets for Polish speech-to-text

-- Evaluation scripts

-- Leaderboard + challenge call (results coming in late '24)

---

> ### Author Response · Authors · 2024-08-26
> **Thank you for detailed feedback**
>
> Dear Reviewer,
>
> Thank you for your detailed and constructive feedback. Here are my responses to your comments:
>
> **Originality:**
> The BIGOS dataset introduced in this paper refers to BIGOS V2, which expands on the previous V1 dataset reported in the [Annals of Computer Science and Information Systems](https://annals-csis.org/proceedings/2023/drp/pdf/1609.pdf). BIGOS V2 includes 12 datasets and evaluates 25 ASR systems, compared to V1’s 10 datasets and 7 systems. Additionally, this work reports results on a conversational dataset curated from PELCRA resources for the first time. The earlier work will be properly cited and compared to clarify these distinctions. An overview of existing benchmarks and a comparison with other Polish benchmarks—including dataset types, numbers, metrics used, and the number of evaluated systems—will be included in the appendix of the revised manuscript. Potential future research direction is extension to other languages of the region, hence brief overview with similar work for other CEE languages will be performed as well.
>
> **Model Benchmarking:**
> The benchmarking section will be expanded with additional details, including definitions and context for metrics (SER, WER, etc.) in Section 2.4.2. Citations for models such as Google STT, Azure STT, and Whisper will also be included. The code used for evaluation, data curation, and visualization is available. Since only existing systems without fine-tuning were considered, code for training will not be included.
>
> **Figures and Plot Generation:**
> Figures 3 and 4 will be revised for clarity. The specific script used for generating these plots will be referenced in the final manuscript.
>
> **Results Analysis:**
> The value of this work lies in the comprehensive evaluation of existing systems across a wide range of scenarios and datasets. The reporting and analysis are designed to be objective and, in my opinion, sufficient given the scope and purpose of the study. Speculation regarding system performance, is intentionally avoided due to limited insight into how systems are developed or fine-tuned, as no additional experiments or fine-tuning were conducted. A comparison between off-the-shelf and fine-tuned systems is planned following the completion of the open challenge for the community. Also a qualitative error analysis to assess error types is planned.
>
> **Documentation:**
> The GitHub repository [link](https://github.com/goodmike31/pl-asr-speech-data-survey) will be updated to include a license and maintenance plan.
>
> Thank you again for your valuable feedback.
>
> Best regards,
> MJ

---

> > ### Comment · Reviewer_j19w · 2024-08-29
> > **Updated score**
> >
> > Thank you for the detailed response and careful consideration of the comments.
> >
> > I have increased the score (5 --> 7) assuming the changes to the paper / Github are made prior to the paper's final submission.

---

### Official Review · Reviewer_h4WC · 2024-07-25
**A large scale evaluation of models specifically for Polish.**

**Rating:** 7
**Confidence:** 4

**Review:**

While the paper has some flaws discussed below, its comprehensiveness means that it will serve as a good guide for how to make large scale speech recognition datasets, as well as for how to combine datasets together. I think it is a very valuable resource for researchers and practitioners who want to improve in this area.

**Strengths:**

The main advantage of this is its comprehensiveness and the ability for it to act as a template for other such works in the future.

**Additional Feedback:**

N/A

**Clarity:**

Normalization is mentioned as important, but it is not clear to me whether normalization is applied to both the reference text as well as the model output transcripts. Some models do normalization, while others don't, so this is an important thing to clarify. It may also affect the correctness of the results.

The subscripts on page 3, at least in my pdf viewer do not appear highlighted as URLs.

**Correctness:**

There is no description of efforts on deduplication of data or verifying whether different datasets contain the same content. This is a common concern for fairness of evaluations, since you don't want the train and test sets to leak into each other. It is straightforward to do such checks using locality sensitive hashing these days.

**Documentation:**

Documentation seems sufficient, but I did not look deeply at the code provided.

**Ethics:**

No.

**Limitations:**

We don't know for sure whether these models were possibly trained on data that also exists in the test sets. This could put at risk many of the claims.

The metrics used for evaluation are limited to WER. For example, there is no work on evaluating how good models are at recognizing specific proper nouns. Speed of transcription also is not evaluated. Model size in terms of parameters is not a good substitute for how fast or slow a model is.

**Opportunities For Improvement:**

I am surprised that OWSM was not used. It seems better than Whisper for academic settings because we know its training data precisely.

Usage of Quartznet nemo models is surprising given their age. A newer model called STT Pl FastConformer Hybrid Transducer-CTC Large P&C is out right now.

**Relation To Prior Work:**

It seems clear this contribution is unique and comprehensive in a way that prior work has not touched upon.

**Summary And Contributions:**

This paper comprehensively describes the way that a large set of datasets were gathered, combined together, and used to evaluate several models for Polish speech recognition.

---

> ### Author Response · Authors · 2024-08-26
> **Thank you for constructive feedback**
>
> Dear Reviewer,
>
> Thank you for your detailed and constructive feedback. I appreciate the time and effort you invested in reviewing the paper. Here are my responses to your suggestions:
>
> **Lack of OWSM Evaluation:**
> Whisper was selected due to its widespread adoption. However, extending the benchmark to include [OWSM 3.1](https://huggingface.co/espnet/owsm_v3.1_ebf) and updating the manuscript is a valuable suggestion. Thank you for this input.
>
> **Nemo Models Coverage:**
> The Quartznet model was included to ensure the benchmark’s comprehensiveness and utility, particularly for practitioners interested in comparing its performance to newer NVIDIA models. Both newer models available for Polish are already included in the benchmark:
> - [STT_pl_fastconformer_hybrid_large_pc](https://huggingface.co/nvidia/stt_pl_fastconformer_hybrid_large_pc) under the codename "nemo_pl_conformer"
> - [STT_multilingual_fastconformer_hybrid_large_pc](https://catalog.ngc.nvidia.com/orgs/nvidia/teams/nemo/models/stt_multilingual_fastconformer_hybrid_large_pc) under the codename "nemo_multilang"
>
> **Risk of Data Leakage:**
> Although deduplication was performed during curation, I will provide a clearer explanation of these efforts. Additionally, the potential limitation of results' reliability, in cases where public recordings used as test data were also used for training the evaluated systems, will be highlighted.
>
> **Evaluation Metrics:**
> Expanding beyond WER to include proper noun recognition, transcription speed, and other metrics is an excellent suggestion. These aspects will be explored in future work to provide a more comprehensive evaluation.
>
> **Other:**
> Thank you for pointing out that the application of normalization to both reference texts and system outputs may not be clear enough, despite being mentioned in Figure 2 and the caption of Table 3. This clarification, along with the issue of subscript visibility on page 3, will be addressed in the final version.
>
> Thank you again for your valuable insights, which will help strengthen the paper.
>
> Best regards,
> MJ

---

### Official Review · Reviewer_VBUM · 2024-08-04
**A very useful benchmark for Polish ASR systems**

**Rating:** 7
**Confidence:** 4

**Review:**

The article presents a comprehensive benchmark consisting of speech datasets, enabling the reproducible evaluation of automatic speech recognition (ASR) systems for Polish. The evaluation is performed over 24 datasets and  25 combinations of ASR systems and models. This study represents an extensive comparison  of commercial and free ASR systems for Polish. Conclusions are drawn from 600 evaluations of system-model-test set combinations, marking a significant advancement in both scale and complexity. Survey results and performance comparisons are available in the form of interactive panels, along with datasets and information about an open challenge. The tools used for evaluation are available as open-source software, facilitating their replication and adaptation to other languages, as well as continuous extension with new datasets and systems. The article addresses the crucial issue of underutilization and lack of standardization of speech datasets, which impedes the development and objective evaluation of ASR systems, particularly for resource-limited languages like Polish.

Despite tackling an important issue, some aspects require improvement:

- the article lacks a detailed description of the criteria for selecting the datasets for analysis and comparison. It is unclear how the authors ensured the representativeness and diversity of the chosen datasets.
- the authors focus mainly on the quantitative evaluation of ASR systems, overlooking a detailed analysis of the types of errors made. A deeper analysis is necessary to understand the causes of differences in results between systems and identify areas that need improvement. Also, this would allow to identify possible problems with the datasets included in the benchmark, e.g., if all evaluated models struggled with a subset of test data points.
- the authors list the 24 datasets used to create BIGOS but do not explain why these particular datasets were chosen. Are there others that could have been included? The article lacks a quantitative analysis of the collected datasets' characteristics. Only general information about the types of speech is provided; more detailed data such as gender, age, dialects, and accents distribution in each dataset are missing.
- the authors mention differences in transcription quality across datasets but do not present any method to evaluate the quality of these transcriptions.

Lastly, one thing that I find unnecessary  is the authors' claim that the paper introduces a "framework" and that Polish is merely a "use case". I really fail to see the "framework" in the paper, it is a very standard pipeline of preparing a benchmark: make literature queries, look for available datasets, check permissions, think about ethical challenges, unify formats, and write code to run tests on evaluated models. I would strongly suggest to the authors to stop over-selling the contribution as a "framework" and simply state the obvious: this is a comprehensive ASR benchmark for Polish ASR. This contribution is really enough.

**Strengths:**

The paper is well written and the contribution is significant. Most importantly, I believe that this is a very useful resource for people working on spoken language in Polish. Given the fact that this is a language natively spoken by 50 mln people, this benchmark can be really very useful. The paper fills a research gap by providing  tools for curating data and evaluating ASR for Polish. The experiments reported in the paper show that an extensive study was conducted, encompassing a large number of datasets and ASR systems, using standardized assessment methods. The work contributes to the advancement of ASR technology for Polish, which has implications for the accessibility of technology for Polish speakers.

**Additional Feedback:**

When reading the paper, a few questions came to mind. Below I am posting these questions hoping that the authors will find them useful when preparing the revision of the paper:

- Is there a plan to add manual transcriptions and annotations to parts of the data for quality verification?
- Is the inclusion of semantic metrics in the evaluation of ASR systems being considered?
- How were the datasets verified for potentially offensive content?
- How was consent obtained from individuals whose recordings are included in datasets where the license allowed it?

In summary, I think that the authors have done significant work in creating a valuable resource for the speech recognition community for the Polish language. The work is well-documented, and the results are discussed in detail. I am sure that many people will find this resource very useful.

**Clarity:**

The article is well-written and organized. The data selection process is clearly described, the results are discussed in detail, and the Appendix presents detailed results. The code, data, and instructions for reproducing the results are made available.

**Correctness:**

The dataset construction appears to be correct. The authors selected and described 24 publicly available datasets and then standardized them. The metrics utilized (SER, WER, MER, WIL, CER) are commonly employed in the evaluation of ASR systems. The experiment design appears to be appropriate. The authors tested 10 ASR systems and 25 models in various scenarios, considering the impact of normalization and factors such as model size and speech rate. I am a bit surprised to see the results of commercial ASR systems (Google, Amazon), I was under impression that the licences of these systems prohibited their use in benchmarks, but I may be wrong.

**Documentation:**

The paper provides detailed information about the origin of the data used to create both BIGOS and PELCRA datasets. The licenses, types of speech (read, spontaneous), recording environments, and details about the speakers are described. The paper describes the process of data categorization, creation of training, validation, and test sets, as well as the standardization of audio file formats and transcriptions. The links to repositories on platforms like Hugging Face and GitHub, where the data, code, and instructions reside, are included in the text. The authors mention obtaining permission from the authors of the PELCRA corpus for its use in benchmarking. They also declare the absence of personal data in the datasets. However, there is a lack of information on the verification of datasets for potentially offensive content.

**Ethics:**

Aside from the mentioned lack of information about verifying datasets for potentially offensive content, there is also an absence of details on how consent was obtained from individuals whose recordings are included in datasets where the license allowed it (e.g., Common Voice).

**Limitations:**

The authors adequately address the limitations of their work and potential negative social impacts, listing them in the "Limitations and Challenges" section. Among other, they list the following limitations:
- lack of manually performed transcriptions and annotations, which hinders the assessment of the quality of test data.
- limited scope of transcription normalization.
- lack of support for embedding-based metrics.
- absence of manual error analysis for speech recognition.
- limited availability of recordings with speaker metadata.
- data sets do not encompass all Polish speakers or the diverse conditions in which ASR systems are used.

This list of limitations is honest and aligns with the main criticisms raised in this review.

**Opportunities For Improvement:**

Below are some suggestions on how the benchmark could be improved:
- the authors could expand the description of the dataset selection criteria and consider the limitations resulting from data availability.
- the authors might provide a detailed analysis of the datasets' characteristics, considering aspects such as: number of hours, number of speakers, gender and age distribution of speakers, types of noise and acoustic conditions
- this might be out of scope of this work, but  a method to assess the quality of transcriptions in the datasets would be very helpful.
- using the results of the evaluation, the authors could conduct a qualitative error analysis, e.g., through manual verification of transcriptions and identification of typical errors made by ASR systems for Polish and investigate the impact of the Polish language's specifics on speech recognition results.

**Relation To Prior Work:**

The authors clearly identify the research gap their work addresses. They note the inadequate use of existing speech datasets for Polish due to issues with their availability, licensing, and interoperability. They also emphasize the lack of a standardized benchmark dataset for Polish, which hampers the development and reliable assessment of ASR systems. The authors reference works that highlight the need for standardized evaluation methodologies in the field of ASR and present arguments for creating improved benchmarks and datasets.

**Summary And Contributions:**

The article presents a comprehensive benchmark of speech datasets and the evaluation of automatic speech recognition (ASR) systems for  Polish. The authors conducted a case study in which the benchmark was utilized to evaluate over 24 datasets and compare 25 combinations of ASR systems and models.

Main contributions include:
- an organized catalog of publicly available speech datasets for the Polish language.
- developing a reference dataset, BIGOS, to serve as a benchmark for evaluating ASR systems.
- a comprehensive evaluation of available ASR systems for Polish using BIGOS and other datasets.
- an interactive dashboards with evaluation results allowing many different dimensions of speech and models.

---

> ### Author Response · Authors · 2024-08-26
> **Thank you for the insightful feedback.**
>
> Dear Reviewer,
>
> Thank you for the detailed and insightful feedback. The time and effort invested in reviewing this work are greatly appreciated. Please see my answers below.
>
> Enhanced Dataset Description: In the revised manuscript, a clearer rationale for the selection made will be provided, along with a more detailed analysis of dataset characteristics, including speaker demographics etc. It should be noted that Polish ASR speech datasets analysis relying on original metadata, as well as automatic content inspection, was already performed prior to submission. This can be referenced in the [BIGOS dataset cards](https://huggingface.co/spaces/amu-cai/amu-bigos-data-dash) and the Polish Speech datasets survey ([1](https://huggingface.co/spaces/amu-cai/pl-asr-survey), [2](https://www.degruyter.com/document/doi/10.1515/psicl-2023-0019/html)).
>
> Qualitative Error Analysis: A qualitative error analysis would indeed be beneficial for assessing error types and their causes, as well as for identifying issues with the test data itself. Such an analysis, at least on a sample of results, is feasible and will be included in the revised manuscript.
>
> Transcription Quality Evaluation: The importance of evaluating transcription quality is recognized. Manual transcription using standard protocol and annotation is planned for future releases. The current approach to improve and assess transcription quality such as manual inspection of  evaluation results and data outliers will be included.
>
> Clarification on the "Framework": Indeed, the terminology in the paper should be adjusted, more accurately presenting the contribution as a comprehensive ASR benchmark specifically for Polish ASR, rather than a generalized "framework". The term will be reconsidered when more systems, datasets, and languages are included in the benchmark and/or tooling is successfully adopted for similar projects by other contributors.
>
> Ethical Considerations: More detailed information will be provided on how consent was obtained in the original datasets, and available methods for detecting potentially offensive content will be explored. This will be addressed in the revised manuscript.
>
> Once again, your constructive feedback is greatly appreciated.
>
> Best regards,
> MJ

---

### Decision · Program_Chairs · 2024-09-26

**Decision:**

Accept (Poster)

**Comment:**

**Summary**

As one of the reviewers did very in depth review and reflects the main paper summary, I am just citing it here as totally agree with it after reading the paper myself:

*The article presents a comprehensive benchmark of speech datasets and the evaluation of automatic speech recognition (ASR) systems for Polish. The authors conducted a case study in which the benchmark was utilized to evaluate over 24 datasets and compare 25 combinations of ASR systems and models.*
*Main contributions include:*
- *an organized catalog of publicly available speech datasets for the Polish language.*
- *developing a reference dataset, BIGOS, to serve as a benchmark for evaluating ASR systems.*
- *a comprehensive evaluation of available ASR systems for Polish using BIGOS and other datasets.*
- *an interactive dashboards with evaluation results allowing many different dimensions of speech and models.*

(though it is BIGOS v2 as pointed out in the discussion and authors need to clarify this).

**Recommendation**

**Reviewer B4Cv** provided justification for the paper which doesn’t have clear arguments having instructions for the D&B track and also contradicts their own words on “providing extensive performance comparisons and insights”. Moreover **Reviewer B4Cv** later pointed out “I realize that there were aspects of this track that I misunderstood.” and thus **I exclude this review** from the final consideration (replies from authors are valid for the points also).

**All other three reviewers** after rebuttal and discussion are **all for acceptance of the paper**, pointing out mainly importance of building evaluation sets for languages other than English and benchmarking recent ASR systems properly for the variety of speech data for different languages, “its comprehensiveness and the ability for it to act as a template for other such works in the future.“, “I believe that this is a very useful resource for people working on spoken language in Polish.”, “The work is well-documented, and the results are discussed in detail. I am sure that many people will find this resource very useful.”

I also agree with reviewers that the paper is an important milestone on documenting all the steps for searching and creating evaluation sets for diverse speech for other languages in the rise of large multilingual models. Even if many parts of that are standard for curating the dataset in speech, it is highly valuable for Polish language as it is a limited-resource language. Moreover, detailed documentation, evaluation and best practices given in the paper can help for other languages' data curation. Based on that, I believe the paper will be interesting to the community for its practices and highlighting the issue of having diverse evaluation sets for languages other than English and our current bias towards the English language in the speech community. With that **I recommend the paper for acceptance**. The only downside: important missing information (discussed in the reviews and rebuttal) must be added into the  revision, and I rely on authors to keep their promise.